# MATES👬: Model-Aware Data Selection for Efficient Pretraining with Data Influence Models

Zichun Yu    Spandan Das    Chenyan Xiong
School of Computer Science
Carnegie Mellon University
{zichunyu, spandand, cx}@andrew.cmu.edu

## Abstract

Pretraining data selection has the potential to improve language model pretraining efficiency by utilizing higher-quality data from massive web data corpora. Current data selection methods, which rely on either hand-crafted rules or larger reference models, are conducted statically and do not capture the evolving data preferences during pretraining. In this paper, we introduce *model-aware data selection with data influence models (MATES)*, where a data influence model continuously adapts to the evolving data preferences of the pretraining model and then selects the data most effective for the current pretraining progress. Specifically, we collect oracle data influence by locally probing the pretraining model and fine-tune a small data influence model to approximate it accurately. The data influence model then predicts data influence over the whole pretraining corpus and selects the most influential data for the next pretraining stage. Experiments of pretraining 410M and 1B models on the C4 dataset demonstrate that MATES significantly outperforms random data selection on extensive downstream tasks. It doubles the gains achieved by the state-of-the-art data selection approach that leverages larger reference models and reduces the total FLOPs required to reach certain performances by half. Further analyses validate the effectiveness of the locally probed oracle data influence and the approximation with data influence models. Our code is open-sourced at https://github.com/cxcscmu/MATES.

## 1 Introduction

The power of large language models (LLMs) rises with scaling up [7; 23; 58]: pretraining models with more *parameters* on more *data* using more *compute* resources [23; 27]. Among these three aspects of scaling, compute is often the most restrictive factor, as current large-scale pretraining frequently demands millions of GPU hours [2; 8; 58], while the model parameters and the pretraining data amounts are determined based on the pre-allocated compute budget [23; 27].

This provides a unique opportunity to elevate the scaling law of pretraining through data selection since the available data sources, such as the web [42; 55], are orders of magnitude bigger than available compute resources and contain data of varying quality. Recent research has shown that effective data selection can improve the generalization ability of pretrained models [15; 62], enhance scaling efficiency [5], and introduce specialized capabilities [34]. These explorations mainly focus on heuristic-based methods, such as rule-based filtering [49; 50; 55], deduplication [1; 45; 57], proximity to high-quality corpora [17; 33; 61; 64], and prompting LLMs [52; 62]. Despite their success on certain datasets and models, these techniques rely heavily on the static heuristics and often overlook the evolving nature of pretraining [39]. As a result, their performance tends to be limited when applied to the real-world pretraining scenarios. [3; 33].

38th Conference on Neural Information Processing Systems (NeurIPS 2024).

The data preferences of pretraining models dynamically shift as they progress through different stages of pretraining [4; 11; 39; 40]. For instance, Figure 1a illustrates how the data influence measured by the pretraining model evolves at different pretraining steps. As a result, the data quality measurement should also keep pace with the model's evolving data preferences during the pretraining. This leads us to our core research question: *How can we precisely track the data influence with the pretraining model and efficiently select pretraining data based on the acquired influence?*

In this paper, we introduce **M**odel-**A**ware data selection with da**T**a influenc**E** model**S** (MATES), a new pretraining paradigm where pretraining data is selected on-the-fly by a data influence model capturing the ever-changing data preferences of the pretraining model. To track the model's data preferences, we locally probe the oracle data influence by evaluating the pretraining model's perfor-

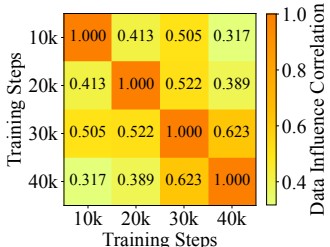
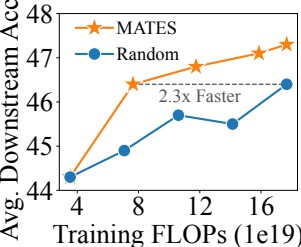

(a) Preference correlation.  (b) Accuracy w.r.t. FLOPs.

Figure 1: Correlation of locally probed data influences at different pretraining steps (a) and the zero-shot performance with model-aware data selection (b). The experiments are based on 1B models.

mance on a reference task after training on individual data points. Then, we train a small data influence model using the locally probed oracle data influence, which then predicts data influence over the whole pretraining corpus and selects the most influential data for the next pretraining stage. This side-by-side learning framework ensures that the data influence model continuously adapts to the evolving data preferences of the pretraining model, providing the most valuable data accordingly.

Our pretraining experiments with 410M and 1B models on the C4 dataset [50] demonstrate that MATES can significantly outperform random selection by an average zero-shot accuracy of 1.3% (410M) and 1.1% (1B) across various downstream tasks, ranging from reading comprehension, commonsense reasoning, and question answering. MATES doubles the gains obtained by state-of-the-art data selection approaches that rely on signals from reference models larger than the pretraining model. Furthermore, our model-aware data selection significantly elevates the scaling curve of pretraining models, as shown in Figure 1b, reducing the total FLOPs required to achieve certain downstream performances by more than half. Further analyses confirm the advantages of our locally probed oracle data influence and the effective approximation of this oracle with data influence models. Ablation studies demonstrate the robustness of MATES across various hyperparameter settings and different design choices of the data influence models.

We summarize our main contributions as follows:

1. We propose a model-aware data selection framework, MATES, where a small data influence model continuously adapts to the constantly changing data preferences of the pretraining model and selects the training data to optimize the efficacy of the pretraining process.

2. We effectively collect oracle data influence through local probing with the pretraining model and use a small BERT-base model to approximate it accurately.

3. We empirically verify the superiority of MATES over rule-based, influence-function-based, and LLM-based selection methods and the effectiveness of the probed oracle and its approximation.

## 2 Related work

Early approaches on data selection relied heavily on manual intuitions. For example, T5 [50] first proposed the C4 pipeline, followed by Gopher rules [49], which utilized criteria like document length, mean word length, and the presence of harmful or stop words to curate data. Recent FineWeb dataset [46] further applied quality and repetition filters on top of these basic rules. These rule-based data selection methods have been shown effective as an initial data curation step [46], though manual intuitions may not capture the nuances of models' data preferences [36].

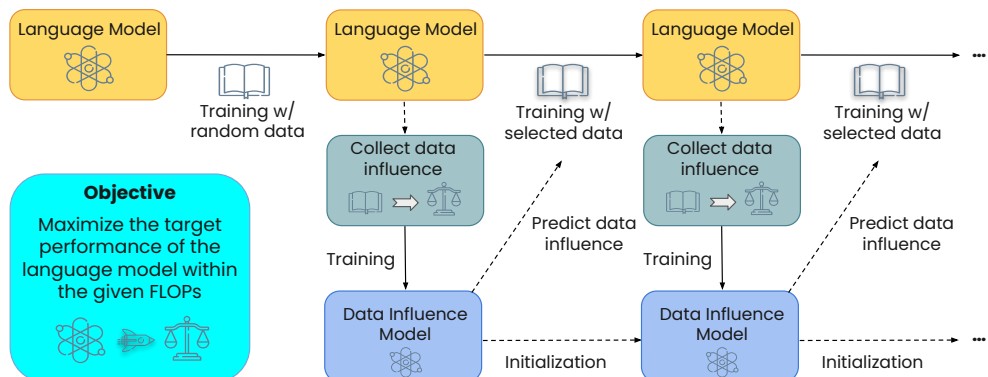

Figure 2: Overview of MATES. The language model is first pretrained with a random set of data. Then, a data influence model is trained to approximate data influences on the target performance of the pretraining model and select the most effective data for the next pretraining stage.

Deduplication is another standard approach in pretraining data selection. Specifically, Lee et al. [32] and Penedo et al. [45] explored exact string match and fuzzy MinHash to filter out duplicate sequences. SemDeDup [1] further leveraged pretrained embeddings to identify semantic duplicates, and D4 [57] introduced diversification factors into the deduplication process. These methods effectively narrow down the number of similar documents in a corpus and are often used together with quality-oriented selection techniques to improve performance.

Selecting data that is proximate to high-quality corpora can enhance the data quality as well [17; 61]. Techniques leveraging n-gram similarity [17; 64; 33] and language modeling perplexity [8; 14; 16; 61] have been adopted to evaluate how closely sequences in large datasets approximate high-quality data. As the size of pretraining data grows, the effectiveness of proximity-based methods becomes unclear, as they potentially reduce the diversity of the pretraining data and, consequently, the general capability of the pretrained models [36].

Recent advancements explore the use of LLMs to improve the pretraining data quality. For instance, QuRating [62] and Ask-LLM [52] employed LLMs like GPT-3.5 to annotate high-quality documents. Maini et al. [38] rephrased web corpora by providing LLMs with detailed prompts to balance quality and diversity. These methods leverage the capabilities of strong reference LLMs, which are often several orders of magnitude larger, to guide the pretraining of smaller models.

Influence functions [29; 59] provide a theoretical tool to assess the impact of individual data points on a model's performance. However, they face scalability challenges in the context of LLMs due to the expensive gradient calculations [21; 54]. To efficiently and robustly approximate influence functions, TRAK [44] performed Taylor approximation and gradient dimension reduction, making influence computation feasible for pretraining experiments [15]. Nevertheless, the computational cost remains prohibitive for model-aware data selection, which requires tracking the evolving data preferences of pretraining models on the fly.

On the other hand, many researchers have proposed curriculum learning strategies that dynamically adjust the data distribution in the pretraining [11; 24; 47; 51; 56; 62]. ELECTRA-style models [11] incorporated curriculum learning in their pretraining process by synchronously training the model with an auxiliary generator, which provided more and more difficult training signals for the discriminator. This implicit curriculum significantly improved the pretraining efficiency on denoising language models [4; 11; 19; 39; 65]. Other methods have explicitly designed the curriculum for pretraining data selection, such as decreasing gradient norms [56], least certainty [24; 51], and increasing expertise [62], demonstrating the benefits of adapting the pretraining data signal according to the model's ever-changing preferences.

## 3 Methods

This section first introduces the model-aware data selection framework MATES (§ 3.1) and then proposes a local probing technique to collect oracle data influence during pretraining (§ 3.2).

**Algorithm 1** Model-Aware Data Selection

---

**Require:** Training data $\mathcal{D}_t$, hold-out data $\mathcal{D}_h$, reference data $\mathcal{D}_r$, pretraining model $\mathcal{M}$, optimizer $\mathcal{A}$, total training step $T$, selected size $k$, data influence model $\Theta$, sampling temperature $\tau$, update step $U$
    Initialize pretraining model parameters $\mathcal{M}$
    Initialize $S_k^*$ as a randomly sampled size-$k$ subset from $\mathcal{D}_t$
    **for** $t = 1, \ldots, T$ **do**
        **if** $t \mod U = 0$ **then**
            Collect oracle data influence $\{\mathcal{I}_\mathcal{M}(x_i; \mathcal{D}_r) \mid x_i \in \mathcal{D}_h\}$
            Fine-tune data influence model $\Theta$ on $\{(x_i, \mathcal{I}_\mathcal{M}(x_i; \mathcal{D}_r)) \mid x_i \in \mathcal{D}_h\}$
            Select data $S_k^* \leftarrow \{\text{Gumbel-Top-}k(\frac{\Theta(x_i)}{\tau}) \mid x_i \in \mathcal{D}_t\}$
        **end if**
        Sample a batch of data $B^*$ from $S_k^*$
        $\mathcal{M} \leftarrow \mathcal{A}(\mathcal{M}, B^*)$
    **end for**

---

## 3.1 Model-aware data selection framework

MATES selects the most effective data for pretraining a language model, aiming to maximize its final target performance, as illustrated in Figure 2. The target performance here can be evaluated using any downstream task or their combinations. Specifically, we leverage non-evaluation data as a reference for the model's target performance and select the pretraining data according to the reference loss.

Formally, given a size-$n$ pretraining dataset $\mathcal{D}_t$ and the current model state $\mathcal{M}$, in each iteration, the objective of data selection is to find an optimal batch $B^*$ from $\mathcal{D}_t$ to minimize the loss $\mathcal{L}$ over the reference data $\mathcal{D}_r$ after training $\mathcal{M}$ on $B^*$:

$$B^* = \arg\min_B \mathcal{L}(\mathcal{D}_r \mid \mathcal{A}(\mathcal{M}, B)) \tag{1}$$

$$\text{where } \mathcal{L}(\mathcal{D}_r \mid \mathcal{A}(\mathcal{M}, B)) = \mathbb{E}_{(x,y) \sim \mathcal{D}_r} \ell(y \mid x; \mathcal{A}(\mathcal{M}, B)), \tag{2}$$

$$\mathcal{M} \leftarrow \mathcal{A}(\mathcal{M}, B^*), \tag{3}$$

where $\mathcal{A}(\mathcal{M}, B)$ denotes the optimization of model $\mathcal{M}$ on a batch $B$, e.g., one-step training with Adam [28] and $\ell$ denotes the function to compute the model loss on an input-output pair $(x, y)$.

There are two challenges to implement this framework. First, enumerating all possible batches will exponentially increase computational complexity. Second, obtaining oracle data influence for all pretraining data points is challenging. To address these issues, we introduce two techniques: pointwise data influence and data influence parameterization.

**Pointwise Data Influence.** To avoid the computationally intensive task of enumerating all possible batches, a more practical workaround is to decompose the group influence into the pointwise influence [15; 44]. Following previous research [44], we aggregate all the data influences by the summation, assuming that each data point $x_i$ has an independent influence irrespective of the others:

$$\mathcal{L}(\mathcal{D}_r \mid \mathcal{A}(\mathcal{M}, B)) = \sum_{x_i \in B} \mathcal{I}_\mathcal{M}(x_i; \mathcal{D}_r), \tag{4}$$

where $\mathcal{I}_\mathcal{M}$ is the oracle pointwise data influence function based on model state $\mathcal{M}$. $\mathcal{I}_\mathcal{M}$ continuously changes along with the model pretraining.

**Data Influence Parameterization.** Estimating oracle pointwise data influence normally involves gradient-based calculation [21; 29; 44], which is impractical to perform over millions of pretraining examples for every pretraining model state. To make the data influence collection feasible, we propose to collect the oracle data influence on a small hold-out dataset $\mathcal{D}_h$ (sampled from the same distribution as $\mathcal{D}_t$) and fine-tune a small *data influence model* $\Theta$ on $\{(x_i, \mathcal{I}_\mathcal{M}(x_i; \mathcal{D}_r)) \mid x_i \in \mathcal{D}_h\}$ to approximate the oracle. This data influence parameterization process transfers the costly influence computation to the small data influence model's inference.

Then, we obtain the influence prediction $\Theta(x_i)$ with the fine-tuned data influence model over all the training examples $x_i \in \mathcal{D}_t$. For better efficiency, we only asynchronously update the data influence model every $U$ steps with the new oracle $\mathcal{I}_\mathcal{M}$. Therefore, one data influence model checkpoint can select the entire data subset $S_k^*$ for the next $U$ steps of pretraining. The selection process uses the Gumbel-Top-$k$ algorithm [30; 62] to sample data from $\mathcal{D}_t$, with influence scores as weights:

$$S_k^* \leftarrow \{\text{Gumbel-Top-}k(\frac{\Theta(x_i)}{\tau}) \mid x_i \in \mathcal{D}_t\}, \tag{5}$$

where $\tau$ is the sampling temperature. The update step $U$ is chosen to balance the data selection efficiency with the evolving data influence. Empirically, we warm up the model with a randomly sampled size-$k$ subset from $\mathcal{D}_t$ in the initial $U$ steps. The overall pretraining and data selection pipeline of MATES is illustrated in Algorithm 1.

## 3.2 Locally probed oracle data influence

The rest of this section presents our method to estimate the oracle data influence $\mathcal{I}_\mathcal{M}$. We start the derivation from the standard influence functions [29; 59] that quantify the reference loss change if one data point $x_i$ in the training data is upweighted by a small $\epsilon$. We denote the optimal model state after the upweighting as $\mathcal{M}_{\epsilon,x_i}^* = \arg\min_\mathcal{M} \frac{1}{n} \sum_{j=1}^n \mathcal{L}(x_j \mid \mathcal{M}) + \epsilon \mathcal{L}(x_i \mid \mathcal{M})$ and simplify the optimal model under $\epsilon = 0$ case (i.e., no upweighting) as $\mathcal{M}^*$. Then, the oracle data influence of upweighting $x_i$ is given by:

$$\mathcal{I}_{\mathcal{M}^*}(x_i; \mathcal{D}_r) \stackrel{\text{def}}{=} \frac{d\mathcal{L}(\mathcal{D}_r \mid \mathcal{M}_{\epsilon,x_i}^*)}{d\epsilon}\bigg|_{\epsilon=0} \tag{6}$$

$$= \nabla_\mathcal{M} \mathcal{L}(\mathcal{D}_r \mid \mathcal{M}^*)^\top \frac{d\mathcal{M}_{\epsilon,x_i}^*}{d\epsilon}\bigg|_{\epsilon=0} \tag{7}$$

$$= -\nabla_\mathcal{M} \mathcal{L}(\mathcal{D}_r \mid \mathcal{M}^*)^\top H_{\mathcal{M}^*}^{-1} \nabla_\mathcal{M} \mathcal{L}(x_i \mid \mathcal{M}^*), \tag{8}$$

where $H_{\mathcal{M}^*} = \frac{1}{n} \sum_{j=1}^n \nabla_\mathcal{M}^2 \mathcal{L}(x_j \mid \mathcal{M}^*)$ is the Hessian and is positive definite. The derivation from Eq. 7 to Eq. 8 is given by building a quadratic approximation to the empirical risk around $\mathcal{M}^*$ and taking a Newton step [29]. Now consider the case that we incorporate $x_i$ into the training data, which means $\epsilon = \frac{1}{n}$, then the parameter change due to the inclusion of $x_i$ is $\mathcal{M}_{\frac{1}{n},x_i}^* - \mathcal{M}^* \approx -\frac{1}{n} H_{\mathcal{M}^*}^{-1} \nabla_\mathcal{M} \mathcal{L}(x_i \mid \mathcal{M}^*)$ and the influence in Eq. 8 can be further represented as:

$$\mathcal{I}_{\mathcal{M}^*}(x_i; \mathcal{D}_r) \approx n \nabla_\mathcal{M} \mathcal{L}(\mathcal{D}_r \mid \mathcal{M}^*)^\top (\mathcal{M}_{\frac{1}{n},x_i}^* - \mathcal{M}^*) \tag{9}$$

$$\approx n(\mathcal{L}(\mathcal{D}_r \mid \mathcal{M}_{\frac{1}{n},x_i}^*) - \mathcal{L}(\mathcal{D}_r \mid \mathcal{M}^*)) \tag{10}$$

$$\propto -\mathcal{L}(\mathcal{D}_r \mid \mathcal{M}^*) + \mathcal{L}(\mathcal{D}_r \mid \mathcal{M}_{\frac{1}{n},x_i}^*). \tag{11}$$

In practice, we manage to obtain the data influence based on the current model state $\mathcal{M}$, while the above influence calculation still remains meaningful in the non-converged state by adding a damping term $\lambda$ that ensures $H_\mathcal{M} + \lambda I$ is positive definite [29]. Under this assumption, the first term in Eq. 11 can be regarded as a fixed value whatever $x_i$ is, since $x_i$ is sampled from the hold-out data $\mathcal{D}_h$. The second term in Eq. 11 can be locally probed with the one-step training of the current model with the new $x_i$, i.e., $\mathcal{A}(\mathcal{M}, x_i))$. This one-step training incorporates $x_i$ into the optimization of the current model. Finally, the influence of $x_i$ on the reference loss is:

$$\mathcal{I}_\mathcal{M}(x_i; \mathcal{D}_r) \propto -\mathcal{L}(\mathcal{D}_r \mid \mathcal{M}) + \mathcal{L}(\mathcal{D}_r \mid \mathcal{A}(\mathcal{M}, x_i)). \tag{12}$$

This formula means, for each $x_i$ in the hold-out data $\mathcal{D}_h$, we run one-step training with the current model $\mathcal{M}$ and evaluate the difference in reference loss before and after one-step training. To ensure that a positive influence score reflects a beneficial impact on model performance, we empirically define the negative influence, $\mathcal{L}(\mathcal{D}_r \mid \mathcal{M}) - \mathcal{L}(\mathcal{D}_r \mid \mathcal{A}(\mathcal{M}, x_i))$, as our locally probed oracle data influence of $x_i$. A full derivation of locally probed oracle data influence can be found in Appendix A.

Table 1: Zero-shot evaluation of pretraining 410M/1B models with different data selection methods. We report the accuracy$_{\text{(standard error)}}$ and the total GPU FLOPs for each method. Dependencies on stronger reference models (e.g., GPT-3.5) are denoted by $*$. Best performances are marked **bold**.

| Methods $_{(\text{\#FLOPs} *1e19)}$ | SciQ | ARC-E | ARC-C | LogiQA | OBQA | BoolQ | HellaSwag | PIQA | WinoGrande | Average |
|---|---|---|---|---|---|---|---|---|---|---|
| **410M Setting:** 410M model, 25B tokens | | | | | | | | | | |
| Random $_{(6.35)}$ | $64.1_{(1.5)}$ | $40.2_{(1.0)}$ | $\mathbf{25.6}_{(1.3)}$ | $24.7_{(1.7)}$ | $29.4_{(2.0)}$ | $58.9_{(0.9)}$ | $39.7_{(0.5)}$ | $67.1_{(1.1)}$ | $50.6_{(1.4)}$ | $44.5_{(1.3)}$ |
| DSIR $_{(6.35)}$ | $63.1_{(1.5)}$ | $39.9_{(1.0)}$ | $23.8_{(1.2)}$ | $27.0_{(1.7)}$ | $28.4_{(2.0)}$ | $58.3_{(0.9)}$ | $39.6_{(0.5)}$ | $66.8_{(1.1)}$ | $51.5_{(1.4)}$ | $44.3_{(1.3)}$ |
| LESS $_{(246.35)}$ | $64.6_{(1.5)}$ | $42.3_{(1.0)}$ | $23.1_{(1.2)}$ | $25.2_{(1.7)}$ | $30.4_{(2.1)}$ | $55.6_{(0.9)}$ | $\mathbf{41.9}_{(0.5)}$ | $67.2_{(1.1)}$ | $51.0_{(1.4)}$ | $44.6_{(1.4)}$ |
| SemDeDup $_{(7.81)}$ | $63.5_{(1.5)}$ | $\mathbf{42.4}_{(1.0)}$ | $24.4_{(1.3)}$ | $\mathbf{27.6}_{(1.8)}$ | $30.0_{(2.1)}$ | $58.2_{(0.9)}$ | $40.8_{(0.5)}$ | $67.8_{(1.1)}$ | $52.3_{(1.4)}$ | $45.2_{(1.4)}$ |
| DsDm $_{(10.72)}$ | $65.4_{(1.5)}$ | $41.7_{(1.0)}$ | $24.7_{(1.3)}$ | $27.5_{(1.8)}$ | $29.0_{(2.0)}$ | $57.5_{(0.9)}$ | $40.3_{(0.5)}$ | $68.1_{(1.1)}$ | $50.1_{(1.4)}$ | $44.9_{(1.4)}$ |
| QuRating$^*$ $_{(26.35)}$ | $64.8_{(1.5)}$ | $42.0_{(1.0)}$ | $25.4_{(1.3)}$ | $25.3_{(1.7)}$ | $30.2_{(2.1)}$ | $58.9_{(0.9)}$ | $40.7_{(0.5)}$ | $67.5_{(1.1)}$ | $52.1_{(1.4)}$ | $45.2_{(1.4)}$ |
| MATES (Ours) $_{(8.11)}$ | $\mathbf{66.0}_{(1.5)}$ | $41.8_{(1.0)}$ | $25.0_{(1.3)}$ | $25.7_{(1.7)}$ | $\mathbf{30.8}_{(2.1)}$ | $\mathbf{60.6}_{(0.9)}$ | $41.0_{(0.5)}$ | $\mathbf{68.7}_{(1.1)}$ | $\mathbf{52.7}_{(1.4)}$ | $\mathbf{45.8}_{(1.4)}$ |
| **1B Setting:** 1B model, 25B tokens | | | | | | | | | | |
| Random $_{(17.67)}$ | $65.8_{(1.5)}$ | $43.7_{(1.0)}$ | $25.6_{(1.3)}$ | $27.5_{(1.8)}$ | $31.8_{(2.1)}$ | $60.2_{(0.9)}$ | $43.8_{(0.5)}$ | $68.9_{(1.1)}$ | $50.7_{(1.4)}$ | $46.4_{(1.4)}$ |
| DSIR $_{(17.67)}$ | $65.8_{(1.5)}$ | $42.6_{(1.0)}$ | $24.7_{(1.3)}$ | $\mathbf{28.7}_{(1.8)}$ | $29.2_{(2.0)}$ | $59.7_{(0.9)}$ | $44.2_{(0.5)}$ | $68.3_{(1.1)}$ | $\mathbf{53.2}_{(1.4)}$ | $46.3_{(1.4)}$ |
| SemDeDup $_{(19.13)}$ | $66.8_{(1.5)}$ | $\mathbf{45.5}_{(1.0)}$ | $25.3_{(1.3)}$ | $27.6_{(1.8)}$ | $30.6_{(2.1)}$ | $60.2_{(0.9)}$ | $45.3_{(0.5)}$ | $69.7_{(1.1)}$ | $52.5_{(1.4)}$ | $47.1_{(1.4)}$ |
| DsDm $_{(22.04)}$ | $\mathbf{68.2}_{(1.5)}$ | $45.0_{(1.0)}$ | $\mathbf{26.5}_{(1.3)}$ | $26.6_{(1.7)}$ | $29.4_{(2.0)}$ | $59.0_{(0.9)}$ | $44.8_{(0.5)}$ | $68.9_{(1.1)}$ | $51.9_{(1.4)}$ | $46.7_{(1.3)}$ |
| QuRating$^*$ $_{(37.67)}$ | $67.1_{(1.5)}$ | $\mathbf{45.5}_{(1.0)}$ | $25.6_{(1.3)}$ | $26.9_{(1.7)}$ | $29.8_{(2.0)}$ | $60.3_{(0.9)}$ | $45.2_{(0.5)}$ | $\mathbf{70.2}_{(1.1)}$ | $51.6_{(1.4)}$ | $46.9_{(1.3)}$ |
| MATES (Ours) $_{(19.97)}$ | $67.3_{(1.5)}$ | $44.9_{(1.0)}$ | $25.9_{(1.3)}$ | $\mathbf{28.7}_{(1.8)}$ | $\mathbf{32.2}_{(2.1)}$ | $\mathbf{60.9}_{(0.9)}$ | $45.3_{(0.5)}$ | $69.5_{(1.1)}$ | $52.4_{(1.4)}$ | $\mathbf{47.5}_{(1.4)}$ |

# 4   Experimental methodologies

**Implementation Details.** We pretrain 410M/1B models with Pythia [5] architecture from scratch on the C4 dataset [50] as our pretraining model $\mathcal{M}$, and continuously fine-tune BERT-base [13] as our data influence model $\Theta$. The data influence model is smaller than the pretraining model, ensuring the efficiency of data selection. More details of data influence models can be found in Appendix B.3. For model pretraining, we utilize Warmup-Stable-Decay (WSD) scheduler proposed in MiniCPM [25], which offers flexibility in the varying training length [22]. More details of WSD scheduler can be found in Appendix B.2.

For MATES selection, we sample 20% data with their influence scores as weights at each pretraining stage (10k steps). The sampling temperature $\tau$ is set to 1.0 to balance the data quality and the diversity. The update step $U$ is also set to 10k so that the selected data is trained by one epoch. Following DsDm [15], we leverage LAMBADA [43] as our reference data $\mathcal{D}_r$. LAMBADA is a widely-used language modeling task and often serves as a validation task for language model pretraining [7; 8; 23]. A summary of these details can be found in Appendix B.1.

**Evaluation Methods.** We use `lm-evaluation-harness` [18] codebase to perform a holistic evaluation of the pretraining models across 9 downstream tasks, including SciQ [60], ARC-E [12], ARC-C [12], LogiQA [35], OBQA [41], BoolQ [10], HellaSwag [66], PIQA [6], and WinoGrande [53]. These tasks cover the core abilities of the pretrained language model, ranging from reading comprehension, commonsense reasoning, and question answering. We report zero-shot accuracy for BoolQ and WinoGrande; otherwise, normalized zero-shot accuracy. We also report the total GPU FLOPs, including model pretraining and data selection, as both are parts of the scaling formulae [20].

The learning outcome of our data influence model is evaluated by the validation Spearman correlation between its predictions and the oracle data influence on a 10% hold-out validation set. This set is sampled from the collected data-oracle mapping $\{(x_i, \mathcal{I}_{\mathcal{M}}(x_i; \mathcal{D}_r)) \mid x_i \in \mathcal{D}_h\}$.

**Baselines.** We compare MATES with random selection as well as the state-of-the-art pretraining data selection baselines, which include (1) DSIR [64]: proximity with Wikipedia by n-gram features. (2) SemDeDup [1]: removal of semantically duplicate sentences. (3) LESS [63]: computing cosine similarity between training and validation (LAMBADA) gradients as influential scores. [1] (4) DsDm [15]: static approximation of influence scores on LAMBADA by a well-trained proxy model [48]. (5) QuRating [62]: ranking with Llama-learned [58] quality scores identified by GPT-3.5 in terms of educational values. These baselines cover all the mainstream data selection schemes, ranging from simple heuristics, static influence functions, and LLM rating. Some recent methods, such as Ask-LLM [52], are not open-sourced yet, prohibiting direct comparisons.

---

[1]Due to the high computational cost of LESS to compute the gradients for all the pretraining examples, we only report its results in the 410M setting.

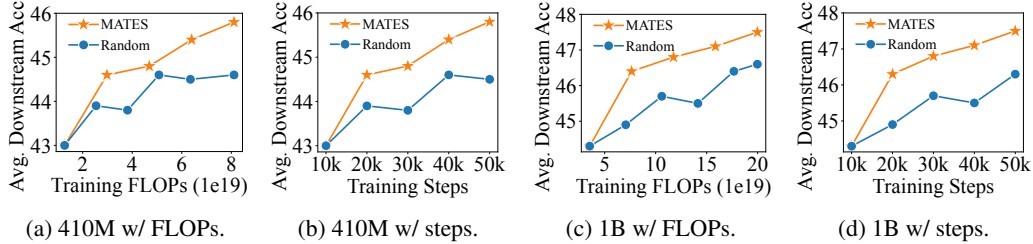

| (a) 410M w/ FLOPs. | (b) 410M w/ steps. | (c) 1B w/ FLOPs. | (d) 1B w/ steps. |

Figure 3: Downstream performance of 410M and 1B models w.r.t. pretraining FLOPs and steps. The data selection procedure of MATES only accounts for 21.7% and 11.5% of the total FLOPs for 410M and 1B models, respectively.

# 5 Evaluation results

This section evaluates the effectiveness of MATES (§ 5.1), locally probed oracle data influence (§ 5.2), and data influence model (§ 5.3). It further presents a case study to illustrate the model's changing data preferences (§ 5.4). More ablation studies can be found in Appendix C.

## 5.1 Overall performance

**MATES outperforms the state-of-the-art data selection approach.** Table 1 presents the zero-shot evaluation of pretraining 410M/1B models with different data selection methods. MATES significantly outperforms random selection by an average downstream accuracy gain of 1.3% and 1.1% in the 410M and 1B settings, respectively. These gains are nearly double those achieved by the state-of-the-art data selection method, QuRating, which depends on larger reference models. Notably, we observe an absolute improvement of nearly 2.0% across most tasks in the 410M setting, except for ARC-C and LogiQA, where the predictions of 410M models are close to random guessing. In the 1B setting, MATES outperforms random selection across all 9 tasks, demonstrating the strong scalability and generalization capabilities of our method.

**MATES selects the data with low costs.** We also show a detailed breakdown of the pretraining cost of MATES in Table 2. The relative selection cost of larger models is generally smaller since their pretraining cost dominates the total FLOPs while the training and inference cost of our data influence model remains stable. Even with the 1B pretraining model, the wall clock time to collect one oracle data influence is only around 2.5 seconds on one GPU, which means we can get all the required 160k oracle scores during 50k-step pretraining on one node (8 GPUs) around 14 hours, which is significantly lower than the actual pretraining time (4 days). The inference speed of our data influence model can also be improved with data parallelism or a fast-inference framework like vLLM [31], re-

Table 2: FLOPs breakdown of MATES steps.

| Process | #FLOPs $*$1e19 | Ratio |
|---|---|---|
| **410M Setting:** 410M model, 25B tokens | | |
| Model pretraining | 6.35 | 78.3% |
| Oracle data influence collection | 0.29 | 3.6% |
| Data influence model training | 0.01 | 0.1% |
| Data influence model inference | 1.46 | 18.0% |
| **Total** | **8.11** | **100%** |
| **1B Setting:** 1B model, 25B tokens | | |
| Model pretraining | 17.67 | 88.5% |
| Oracle data influence collection | 0.83 | 4.1% |
| Data influence model training | 0.01 | 0.1% |
| Data influence model inference | 1.46 | 7.3% |
| **Total** | **19.97** | **100%** |

ducing the selection cost further. This breakdown analysis underscores the low expense of MATES in achieving model-aware data selection.

**MATES significantly elevates the scaling curves.** Figure 3 plots the performance of the pretraining models w.r.t. different FLOPs and steps. Measuring by FLOPs counts both the model pretraining and data selection costs, demonstrating the total compute expense during pretraining. Measuring by steps reflects the compute cost of the model pretraining alone, as the data selection can be trivially parallelized when more computational resources are available. At both 410M and 1B scales, MATES

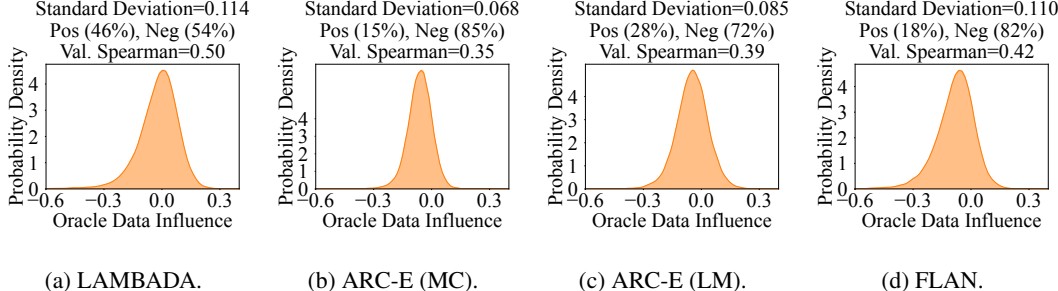

|            | (a) LAMBADA. | (b) ARC-E (MC). | (c) ARC-E (LM). | (d) FLAN. |

Figure 4: Oracle data influence distribution in the 410M setting with different reference tasks at 50k steps. MC: multiple choice. LM: language modeling. We also present the standard deviation of the distribution and the proportions of the data with positive/negative oracle data influence.

Table 3: Performances of oracle selected data with different reference tasks in the 410M setting. We run the decay stage starting from the MATES model at 50k steps.

| $\mathcal{D}_r$ | SciQ | ARC-E | ARC-C | LogiQA | OBQA | BoolQ | HellaSwag | PIQA | WinoGrande | Average |
|---|---|---|---|---|---|---|---|---|---|---|
| LAMBADA | 66.0$_{(1.5)}$ | 42.2$_{(1.0)}$ | 24.8$_{(1.3)}$ | 27.2$_{(1.7)}$ | 30.8$_{(2.1)}$ | **59.1**$_{(0.9)}$ | **41.9**$_{(0.5)}$ | **68.5**$_{(1.1)}$ | 52.3$_{(1.4)}$ | 45.9$_{(1.4)}$ |
| ARC-E (MC) | 64.9$_{(1.5)}$ | 42.4$_{(1.0)}$ | 24.9$_{(1.3)}$ | 27.8$_{(1.8)}$ | 30.4$_{(2.1)}$ | 58.0$_{(0.9)}$ | 41.1$_{(0.5)}$ | 68.1$_{(1.1)}$ | 51.7$_{(1.4)}$ | 45.5$_{(1.4)}$ |
| ARC-E (LM) | 65.3$_{(1.5)}$ | 43.0$_{(1.0)}$ | 24.8$_{(1.3)}$ | 28.0$_{(1.8)}$ | 31.8$_{(2.1)}$ | 58.5$_{(0.9)}$ | 40.7$_{(0.5)}$ | 67.2$_{(1.1)}$ | **52.5**$_{(1.4)}$ | 45.8$_{(1.4)}$ |
| FLAN | **66.4**$_{(1.5)}$ | **45.1**$_{(1.0)}$ | **25.1**$_{(1.3)}$ | **28.7**$_{(1.8)}$ | **32.0**$_{(2.1)}$ | 56.2$_{(0.9)}$ | 40.5$_{(0.5)}$ | 67.9$_{(1.1)}$ | 52.3$_{(1.4)}$ | **46.0**$_{(1.4)}$ |

significantly elevates the scaling curves compared to random selection, reducing the FLOPs and pretraining steps required to reach a certain downstream performance by more than half. Scaling efficiency is more evident at the 1B scale, where model pretraining dominates 88.5% of FLOPs versus 11.5% for data selection. As a result, MATES reaches the same performance as random selection with only 43.3% of the total FLOPs. A similar observation can be found in Figure 9, where the performance of MATES is comparable to or higher than the full pretraining using 3x data. These results reveal the promising potential of MATES in elevating the scaling law of foundation models.

## 5.2 Effectiveness of locally probed oracle data influence

This set of experiments analyzes the effectiveness of oracle data influence with different reference tasks. Besides LAMBADA used in our main experiment, we also consider taking the training sets of ARC-E [12] and FLAN [9] as the reference tasks to show the generalization ability of our method. ARC-E represents one of the knowledge-based question-answering tasks, while FLAN represents a large set of varied instruction-formatted data. For ARC-E, we construct each example either as a multiple-choice selection (i.e., outputting the correct option [A-D]) or as a language modeling task (i.e., outputting the verbalized answer) to investigate whether the task format will affect the data influence collection and parameterization.

In Figure 4, we demonstrate the oracle data influence distribution with different reference tasks, the standard deviation of the distribution, and the proportions of the data with positive/negative oracle data influence. The oracle distribution remains spread-out across all reference tasks, indicating that our oracle effectively differentiates data influences. Notably, the positive influence proportion for LAMBADA in Figure 4a is higher than others by more than 20%, suggesting that more data is deemed beneficial when LAMBADA serves as the reference. We hypothesize that this is because LAMBADA is essentially a word prediction task, which aligns more closely with the pretraining objective compared to knowledge-based or instruction-following tasks. This hypothesis is further supported by the results for ARC-E (Figure 4b/4c), where the language modeling format identifies more positively influential data points compared to the multiple-choice format.

We further validate the effectiveness of the oracle across different reference tasks by sampling 20% data with oracle scores as weights. Due to the infeasibility of obtaining oracle scores for all the pretraining data, we only run one short decay stage with the data selected by different oracle scores.

As shown in Table 3, taking ARC-E as the reference task can benefit the model's in-domain accuracy, but its generalization performance is worse than using LAMBADA. In contrast, FLAN benefits a wider range of downstream tasks due to its diverse instructions. However, there remains a trade-off between the performance of different tasks, so the average accuracy of choosing FLAN is similar to that of LAMBADA. This experiment highlights the robustness of our oracle, as it is not limited to specific reference data but generalizes effectively across multiple reference tasks.

## 5.3 Effectiveness of data influence model

This experiment studies the effectiveness of data influence model. As shown in Figure 4, our data influence model effectively approximates the oracle data influence across various reference tasks. Nevertheless, a higher standard deviation in the oracle distribution tends to enhance the validation Spearman correlation. This suggests that greater variability in influence scores can provide more diverse signals for the data influence model to capture, making it better approximate the oracle distribution.

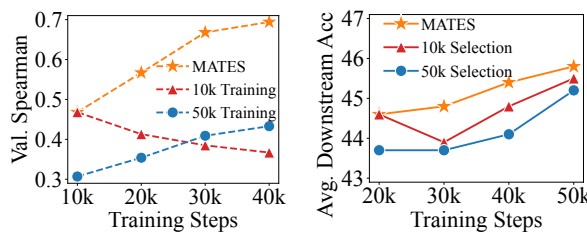

(a) Influence modeling.  (b) Downstream accuracy.

Figure 5: Static (based on a 10k or a 50k random-pretrained model checkpoint) data selection versus model-aware data selection in influence modeling and downstream accuracy.

Figure 5 compares MATES with static data influence models trained on influence from a 10k or a 50k random-pretrained model checkpoint. In Figure 5a, we measure the validation Spearman correlation between the predictions of data influence models and the oracle data influence probed with each pretraining checkpoint. The correlations of static data influence models are always below 0.5, while data influence models in MATES, with dynamic updates, can capture the ever-changing data preferences more and more precisely along with the model pretraining (e.g., the correlation is around 0.7 in 40k steps). The effects of model-aware data selection are directly reflected in downstream accuracy. In Figure 5b, the data selected by static data influence models will cause a notable performance drop, especially at the early pretraining stage. These observations confirm the ever-changing nature of data preferences in pretraining and the advantages of model-aware data selection to elevate the scaling curves of pretraining.

## 5.4 Case study

This case study demonstrates representative examples to illustrate the evolving preferences of the pretraining model in detail. As shown in Table 4, the model at the early pretraining stage (the 10k checkpoint) tends to favor learning natural narrative examples without delving too deeply into specific knowledge. At the 20k checkpoint, it appears to shift focus toward factual knowledge (e.g., pages from Wikipedia) while gradually reducing reliance on natural narratives. At the 30k checkpoint, the model shows a preference for more detailed academic text, such as official teaching slides. By the 40k checkpoint, the model may begin learning more long-tail knowledge, like Telescopic Forklift Training. These examples provide insights into the evolving nature of the model's data preferences throughout pretraining. Although they may not cover every aspect of the selected data, our observation highlights the necessity to adapt data selection strategies to different model learning stages.

## 6 Discussion and limitations

**Combinational Measurement of Data Influence.** One primary assumption in our work is that each data point contributes independently to the pretraining outcome, without accounting for its interactions with other data points. Despite a common hypothesis [15; 44], the pretraining essentially applies the long-term combinational effect of batched data on language models, and the learning of many advanced capabilities is accumulative. Efficiently and effectively measuring and learning the combinational and accumulative nature of the pretraining process can make us better understand and leverage the value of data [37].

Table 4: Case study in the 1B setting. We show the 300 character excerpts of data points with the ranks of their influence scores among 80k randomly sampled data at different pretraining checkpoints. A lower rank denotes a higher influence.

| Influence Rank ↓ | Source | Text |
|---|---|---|
| *0* at *10k* checkpoint
*6376* at *20k* checkpoint | Blog | She went to the place where Jonathan lay and gave to his servant's David's richest garment to be placed next to him as he lay crying out in his sickness. She went in and out of the house. She went in and out of the city gates. She waited for David in the place... |
| *1* at *20k* checkpoint
*22998* at *30k* checkpoint | Wikipedia | Two weeks later, Friedman threw three touchdown passes in a 27–0 victory over Northwestern. One of Michigan's touchdowns was set up when Friedman intercepted a Northwestern pass and returned it 13 yards. On the next play, Friedman threw a touchdown pass to Dutch Marion... |
| *3* at *30k* checkpoint
*452* at *40k* checkpoint | CRITHINKEDU | Critical Thinking Across the European Higher Education Curricula), Education and Regional Development in Southern Europe: Should we invest in Critical Thinking across the Higher Education Curricula? First... What is Critical Thinking (CT)? CT is not only a high quality way of thinking (skill), but also a way of being (disposition)... |
| *1* at *40k* checkpoint
*5954* at *10k* checkpoint | forkliftcertification | It's a group of training course resources to help you master telescopic forklifts in record time. Or else you're taking the course and throwing a bunch of forklift telescopic training against a wall and hoping something sticks. And forklift is only getting more popular. This chapter is about handler course and certification... |

**Exploratory Scale.** As an exploratory research work, our experiments are conducted at a moderate scale, with a pretraining model of 410M or 1B parameters. Although the trend from 410M to 1B indicates the robustness of our observations, it remains unclear how well our methods scale up to production-level models with billions of parameters and trillions of pretraining tokens. On the one hand, moving to that scale provides more headroom for data selection with more urgent needs for efficiency, more leniency on the relatively small compute spent on data selection, and a larger pool of candidate data. On the other hand, large-scale pretraining may yield various stability issues that require dedicated work to introduce new techniques [58; 67]. We leave the exploration of larger models to future work.

## 7   Conclusion

In this paper, we introduce MATES, a novel framework to enhance the efficiency and effectiveness of language model pretraining through model-aware data selection. MATES leverages a data influence model to continuously capture the evolving data preferences of the pretraining model throughout the pretraining process, thereby selecting the training data most effective for the current pretraining stage. To achieve that, we locally probe the oracle data influence on a reference task using the pretraining model and fit the data influence model on the probed oracle. Our empirical results demonstrate that MATES surpasses random, rule-based, influence-function-based, and LLM-based data selection methods on pretraining, significantly elevating the scaling curves of pretraining LLMs. Further analyses confirm the effectiveness of our locally probed oracle data influence and the accurate approximation of this oracle with data influence models. Our work successfully demonstrates the potential of model-aware data curation in pretraining, and we hope it will motivate further explorations on improving the scaling law of foundation models through better data curation techniques.

## Acknowledgements

We sincerely thank Jie Lei, Fei Peng, Scott Yih, and Arnold Overwijk for discussing ideas and providing helpful feedback on this work.

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

# A   Full derivation of locally probed oracle data influence

This section presents a full derivation of locally probed oracle data influence. First, we define the optimal model state after upweighting one training data point $x_i$ by $\epsilon$ as:

$$\mathcal{M}^*_{\epsilon,x_i} = \arg\min_{\mathcal{M}} \frac{1}{n}\sum_{j=1}^{n} \mathcal{L}(x_j \mid \mathcal{M}) + \epsilon\mathcal{L}(x_i \mid \mathcal{M}), \tag{13}$$

and simplify the optimal model under $\epsilon = 0$ case (i.e., no upweighting) as $\mathcal{M}^*$. Following Koh and Liang [29], data influence quantifies the reference loss change before and after one data point $x_i$ in the training data is upweighted by a small $\epsilon$, i.e.:

$$\mathcal{I}_{\mathcal{M}^*}(x_i; \mathcal{D}_r) \overset{\text{def}}{=} \left.\frac{d\mathcal{L}(\mathcal{D}_r \mid \mathcal{M}^*_{\epsilon,x_i})}{d\epsilon}\right|_{\epsilon=0} \tag{14}$$

$$= \nabla_{\mathcal{M}}\mathcal{L}(\mathcal{D}_r \mid \mathcal{M}^*)^\top \left.\frac{d\mathcal{M}^*_{\epsilon,x_i}}{d\epsilon}\right|_{\epsilon=0}. \tag{15}$$

To compute $\left.\frac{d\mathcal{M}^*_{\epsilon,x_i}}{d\epsilon}\right|_{\epsilon=0}$, we consider the optimality condition at $\mathcal{M}^*_{\epsilon,x_i}$:

$$\nabla_{\mathcal{M}}\left[\frac{1}{n}\sum_{j=1}^{n} \mathcal{L}(x_j \mid \mathcal{M}^*_{\epsilon,x_i}) + \epsilon\mathcal{L}(x_i \mid \mathcal{M}^*_{\epsilon,x_i})\right] = 0. \tag{16}$$

Differentiating both sides with respect to $\epsilon$:

$$H_{\mathcal{M}^*} \left.\frac{d\mathcal{M}^*_{\epsilon,x_i}}{d\epsilon}\right|_{\epsilon=0} + \nabla_{\mathcal{M}}\mathcal{L}(x_i \mid \mathcal{M}^*) = 0, \tag{17}$$

$$\left.\frac{d\mathcal{M}^*_{\epsilon,x_i}}{d\epsilon}\right|_{\epsilon=0} = -H_{\mathcal{M}^*}^{-1}\nabla_{\mathcal{M}}\mathcal{L}(x_i \mid \mathcal{M}^*), \tag{18}$$

where $H_{\mathcal{M}^*} = \frac{1}{n}\sum_{j=1}^{n}\nabla_{\mathcal{M}}^2\mathcal{L}(x_j \mid \mathcal{M}^*)$ is the Hessian and is positive definite. Therefore,

$$\mathcal{I}_{\mathcal{M}^*}(x_i; \mathcal{D}_r) = -\nabla_{\mathcal{M}}\mathcal{L}(\mathcal{D}_r \mid \mathcal{M}^*)^\top H_{\mathcal{M}^*}^{-1}\nabla_{\mathcal{M}}\mathcal{L}(x_i \mid \mathcal{M}^*). \tag{19}$$

With the first-order Taylor expansion, we can approximate the change in model parameters:

$$\mathcal{M}^*_{\epsilon,x_i} - \mathcal{M}^* \approx \epsilon \left.\frac{d\mathcal{M}^*_{\epsilon,x_i}}{d\epsilon}\right|_{\epsilon=0} = -\epsilon H_{\mathcal{M}^*}^{-1}\nabla_{\mathcal{M}}\mathcal{L}(x_i \mid \mathcal{M}^*). \tag{20}$$

For $\epsilon = \frac{1}{n}$, i.e., incorporating $x_i$ into the training data, this becomes:

$$\mathcal{M}^*_{\frac{1}{n},x_i} - \mathcal{M}^* \approx -\frac{1}{n}H_{\mathcal{M}^*}^{-1}\nabla_{\mathcal{M}}\mathcal{L}(x_i \mid \mathcal{M}^*). \tag{21}$$

Substituting $H_{\mathcal{M}^*}^{-1}\nabla_{\mathcal{M}}\mathcal{L}(x_i \mid \mathcal{M}^*)$ in Eq. 19 with $-n(\mathcal{M}^*_{\frac{1}{n},x_i} - \mathcal{M}^*)$, we get:

$$\mathcal{I}_{\mathcal{M}^*}(x_i; \mathcal{D}_r) \approx n\nabla_{\mathcal{M}}\mathcal{L}(\mathcal{D}_r \mid \mathcal{M}^*)^\top(\mathcal{M}^*_{\frac{1}{n},x_i} - \mathcal{M}^*) \tag{22}$$

$$\approx n(\mathcal{L}(\mathcal{D}_r \mid \mathcal{M}^*_{\frac{1}{n},x_i}) - \mathcal{L}(\mathcal{D}_r \mid \mathcal{M}^*)) \tag{23}$$

$$\propto -\mathcal{L}(\mathcal{D}_r \mid \mathcal{M}^*) + \mathcal{L}(\mathcal{D}_r \mid \mathcal{M}^*_{\frac{1}{n},x_i}). \tag{24}$$

In practice, we manage to obtain the data influence based on the current model state $\mathcal{M}$, while the above influence calculation still remains meaningful in the non-converged state by adding a damping term $\lambda$ that ensures $H_{\mathcal{M}} + \lambda I$ is positive definite [29]. Finally, the locally probed oracle data influence of $x_i$ is:

$$\mathcal{I}_{\mathcal{M}}(x_i; \mathcal{D}_r) \propto -\mathcal{L}(\mathcal{D}_r \mid \mathcal{M}) + \mathcal{L}(\mathcal{D}_r \mid \mathcal{A}(\mathcal{M}, x_i)). \tag{25}$$

To ensure that a positive influence score reflects a beneficial impact on model performance, we empirically define the negative influence, $\mathcal{L}(\mathcal{D}_r \mid \mathcal{M}) - \mathcal{L}(\mathcal{D}_r \mid \mathcal{A}(\mathcal{M}, x_i))$, as our locally probed oracle data influence of $x_i$.

Table 5: Experimental configurations.

| Configuration Name | Value |
|---|---|
| *Pretraining* | |
| Dataset | C4 |
| Tokens | 25B |
| Model | Pythia-410M/1B (randomly initialized) |
| Steps | 50k |
| Sequence length | 1024 |
| Batch size | 512 |
| Max learning rate | 0.001 |
| *Data influence parameterization* | |
| Amount of collected oracle data | 80k at 10k steps
20k at the following steps |
| Initialization of data influence model | Pretrained BERT at 10k steps
Last fine-tuned checkpoint at the following steps |
| Selection ratio | 20% |
| Update step of data influence model | 10k |
| Amount of data in reference tasks | 1024 |
| Epochs | 5 |
| Batch size | 256 |
| Max learning rate | 5e-5 |
| Validation set of data influence model | 10% sampled from the collected oracle data |

# B  Experimental details

This section provides our experimental configurations (§ B.1), details of WSD scheduler (§ B.2), and design of data influence model (§ B.3).

## B.1  Experimental configurations

We provide all experimental configurations in Table 5.

## B.2  WSD scheduler

WSD learning rate scheduler is initially proposed in MiniCPM [25] and is found to scale predictably and reliably similar to the widely-used cosine learning rate scheduler [22]. Moreover, WSD offers better flexibility than cosine for training across different lengths since its learning rate is constant during the stable stage. The learning rate of WSD scheduler is configured as follows:

$$lr(t) = \begin{cases} \frac{t}{W} \cdot \eta, & \text{if } t < W \\ \eta, & \text{if } W \leq t < S \\ 0.5^{4 \cdot (t-S)/D} \cdot \eta, & \text{if } S \leq t < S + D \end{cases} \quad (26)$$

where $t$, $W$, $S$, and $D$ represent the number of steps now, at the end of the warmup, stable, and decay stages, respectively. $\eta$ is the max learning rate. In our main experiments, we choose $W = 2000$, $S = 50000$, and $D = 200$. Inspired by MiniCPM [25], each checkpoint is evaluated after the short decay stage for better stability. We run all experiments on 8 A6000 GPUs, which will take 2 days for 410M models and 4 days for 1B models.

## B.3  Design of data influence model

Our BERT-based data influence model averages all the hidden representations of the last model layer to obtain the sequence representation $\mathbf{h} \in \mathbb{R}^H$, where $H$ is the hidden size of the model. Note that BERT can only support a maximum input sequence length of 512. To deal with our pretraining sequence length of 1024, we divide one sequence into two chunks and forward them separately. Then, we average the hidden representations from both chunks to obtain the final $\mathbf{h}$. This vector will be multiplied by a regression output weight $\mathbf{w}_o \in \mathbb{R}^H$ to get the

model prediction $\mathbf{w}_o \cdot \mathbf{h}$. The training objective is the mean squared error between the model prediction and the normalized ground truth $\mathcal{I}_{\mathcal{M}}$.

Note that BERT has a different tokenizer from our pretraining models, but the average sequence length of all the examples after the tokenization is almost the same. Our chunk-based design can be easily extended to longer pretraining sequences in the future.

# C  Additional experiments

This section presents the comparison of different data influence attribution methods (§ C.1), the ablation study on data influence models (§ C.2), and the analysis of oracle data influence (§ C.3).

## C.1  Comparison of different data influence attribution methods

Table 6: Performances of locally probed oracle data influence, MATES, and DsDm in 410M setting at 40k steps. We show zero-shot/two-shot results.

| Methods | SciQ | ARC-E | ARC-C | LogiQA | OBQA |
|---|---|---|---|---|---|
| Oracle | $65.4_{(1.5)}/70.4_{(1.4)}$ | $\mathbf{42.5}_{(1.0)}/43.6_{(1.0)}$ | $\mathbf{25.2}_{(1.3)}/25.0_{(1.3)}$ | $26.1_{(1.7)}/25.7_{(1.7)}$ | $\mathbf{31.8}_{(2.1)}/\mathbf{30.4}_{(2.1)}$ |
| MATES | $\mathbf{67.3}_{(1.5)}/\mathbf{76.7}_{(1.3)}$ | $41.7_{(1.0)}/\mathbf{44.4}_{(1.0)}$ | $24.7_{(1.3)}/24.0_{(1.2)}$ | $\mathbf{26.9}_{(1.7)}/\mathbf{26.3}_{(1.7)}$ | $28.8_{(2.0)}/28.0_{(2.0)}$ |
| DsDm | $66.0_{(1.5)}/72.7_{(1.4)}$ | $41.7_{(1.0)}/43.2_{(1.0)}$ | $23.7_{(1.2)}/\mathbf{25.2}_{(1.3)}$ | $24.4_{(1.7)}/23.3_{(1.7)}$ | $29.2_{(2.0)}/29.4_{(2.0)}$ |

| Methods | BoolQ | HellaSwag | PIQA | WinoGrande | Average |
|---|---|---|---|---|---|
| Oracle | $58.9_{(0.9)}/\mathbf{59.1}_{(0.9)}$ | $\mathbf{41.1}_{(0.5)}/\mathbf{43.1}_{(0.5)}$ | $\mathbf{68.2}_{(1.1)}/66.6_{(1.1)}$ | $51.6_{(1.4)}/\mathbf{53.2}_{(1.4)}$ | $\mathbf{45.6}_{(1.4)}/\mathbf{46.3}_{(1.3)}$ |
| MATES | $59.6_{(0.9)}/57.0_{(0.9)}$ | $40.1_{(0.5)}/39.6_{(0.5)}$ | $67.6_{(1.1)}/\mathbf{67.7}_{(1.1)}$ | $\mathbf{52.1}_{(1.4)}/51.3_{(1.4)}$ | $45.4_{(1.3)}/46.1_{(1.3)}$ |
| DsDm | $\mathbf{60.3}_{(0.9)}/58.1_{(0.9)}$ | $40.4_{(0.5)}/40.2_{(0.5)}$ | $67.2_{(1.1)}/66.5_{(1.1)}$ | $50.4_{(1.4)}/52.2_{(1.4)}$ | $44.8_{(1.3)}/45.6_{(1.3)}$ |

We compare the performance of data selection directly using our oracle with DsDm at the short decay stage in Table 6. Our locally probed oracle, utilizing one-step training, outperforms DsDm, which relies on Taylor approximation and gradient dimension reduction. The performance gap (0.8 zero-shot accuracy gain) is significant, considering the decay stage only consists of 200 steps. This improvement can be attributed to two factors: (1) we leverage the current model state to calculate the data influence rather than relying on an existing checkpoint like DsDm, and (2) we perform one-step training to obtain the oracle score, considering the training dynamics of the pretraining model and eliminating the precision loss from multiple approximation manipulations in DsDm. It is always costly to acquire the optimal data influence oracle for the entire pretraining dataset, but our local probing method with data influence parameterization beats Taylor approximation with fewer FLOPs, offering a new direction for future exploration.

## C.2  Ablation study on data influence models

Table 7: Ablation study of pretraining 410M models with different update steps $U$ and sampling temperatures $\tau$ between 40k to 50k steps. We show zero-shot/two-shot results.

| Hyperparameters | SciQ | ARC-E | ARC-C | LogiQA | OBQA |
|---|---|---|---|---|---|
| $U$=10k, $\tau$=1.0 | $\mathbf{66.0}_{(1.5)}/\mathbf{74.9}_{(1.4)}$ | $41.8_{(1.0)}/43.8_{(1.0)}$ | $\mathbf{25.0}_{(1.3)}/\mathbf{25.3}_{(1.3)}$ | $25.7_{(1.7)}/24.9_{(1.7)}$ | $30.8_{(2.1)}/\mathbf{30.6}_{(2.1)}$ |
| $U$=5k, $\tau$=1.0 | $65.8_{(1.5)}/74.6_{(1.4)}$ | $\mathbf{42.2}_{(1.0)}/44.1_{(1.0)}$ | $24.8_{(1.3)}/25.1_{(1.3)}$ | $25.3_{(1.7)}/25.8_{(1.7)}$ | $\mathbf{31.0}_{(2.1)}/30.0_{(2.1)}$ |
| $U$=2.5k, $\tau$=1.0 | $65.0_{(1.5)}/74.4_{(1.4)}$ | $41.8_{(1.0)}/43.5_{(1.0)}$ | $24.9_{(1.3)}/24.7_{(1.3)}$ | $27.6_{(1.8)}/\mathbf{26.4}_{(1.7)}$ | $30.6_{(2.1)}/28.8_{(2.0)}$ |
| $U$=10k, $\tau$=2.0 | $64.0_{(1.5)}/72.2_{(1.4)}$ | $41.8_{(1.0)}/\mathbf{44.3}_{(1.0)}$ | $24.7_{(1.3)}/23.5_{(1.2)}$ | $27.8_{(1.8)}/24.9_{(1.7)}$ | $29.6_{(2.0)}/\mathbf{30.6}_{(2.1)}$ |
| $U$=10k, $\tau$=0.0 | $65.9_{(1.5)}/74.5_{(1.4)}$ | $41.8_{(1.0)}/43.5_{(1.0)}$ | $24.6_{(1.3)}/23.5_{(1.2)}$ | $\mathbf{28.0}_{(1.8)}/24.6_{(1.7)}$ | $30.0_{(2.1)}/28.8_{(2.0)}$ |

| Hyperparameters | BoolQ | HellaSwag | PIQA | WinoGrande | Average |
|---|---|---|---|---|---|
| $U$=10k, $\tau$=1.0 | $\mathbf{60.6}_{(0.9)}/57.4_{(0.9)}$ | $41.0_{(0.5)}/40.6_{(0.5)}$ | $\mathbf{68.7}_{(1.1)}/67.1_{(1.1)}$ | $\mathbf{52.7}_{(1.4)}/\mathbf{53.4}_{(1.4)}$ | $\mathbf{45.8}_{(1.4)}/46.4_{(1.3)}$ |
| $U$=5k, $\tau$=1.0 | $60.3_{(0.9)}/\mathbf{58.8}_{(0.9)}$ | $\mathbf{41.1}_{(0.5)}/\mathbf{40.8}_{(0.5)}$ | $68.2_{(1.1)}/\mathbf{68.1}_{(1.1)}$ | $52.5_{(1.4)}/52.2_{(1.4)}$ | $45.7_{(1.4)}/\mathbf{46.6}_{(1.3)}$ |
| $U$=2.5k, $\tau$=1.0 | $60.2_{(0.9)}/57.5_{(0.9)}$ | $41.0_{(0.5)}/40.6_{(0.5)}$ | $67.5_{(1.1)}/67.1_{(1.1)}$ | $51.5_{(1.4)}/51.1_{(1.4)}$ | $45.6_{(1.4)}/46.0_{(1.3)}$ |
| $U$=10k, $\tau$=2.0 | $56.2_{(0.9)}/54.3_{(0.9)}$ | $40.8_{(0.5)}/40.5_{(0.5)}$ | $67.7_{(1.1)}/67.1_{(1.1)}$ | $52.2_{(1.4)}/51.4_{(1.4)}$ | $45.0_{(1.4)}/45.4_{(1.3)}$ |
| $U$=10k, $\tau$=0.0 | $57.4_{(0.9)}/55.8_{(0.9)}$ | $40.7_{(0.5)}/40.6_{(0.5)}$ | $68.1_{(1.1)}/66.8_{(1.1)}$ | $50.6_{(1.4)}/51.1_{(1.4)}$ | $45.2_{(1.4)}/45.5_{(1.3)}$ |

This group of experiments conducts ablation studies on the key hyperparameters.

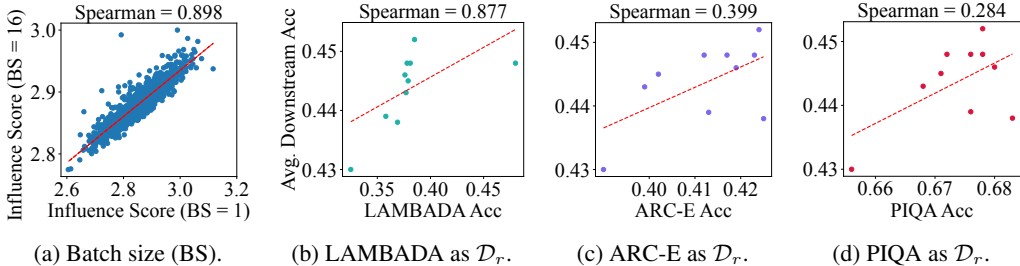

| (a) Batch size (BS). | (b) LAMBADA as $\mathcal{D}_r$. | (c) ARC-E as $\mathcal{D}_r$. | (d) PIQA as $\mathcal{D}_r$. |

Figure 8: Correlation of oracle data influence probed with different batch sizes (a) and the correlation between different reference task accuracy with average downstream accuracy (b-d).

**Update Step and Sampling Temperature.** Table 7 shows the impact of hyperparameters on downstream performance. We initialize the model with the 40k-step MATES checkpoint and adopt different hyperparameters in the 40k-50k training. Decreasing the update step $U$ from 10k to 5k and 2.5k leads to little fluctuations since the model preferences may not dramatically change within 5k steps. Varying the sampling temperature $\tau$ to 2.0 and 0.0 causes decreased performances. The zero temperature extremely up-weights high-quality data, reducing data diversity, while higher temperatures like 2.0 do not sufficiently emphasize data quality. Our observations highlight the necessity of balancing quality and diversity in the data selection.

**Data Selection Ratio.** As shown in Figure 6, MATES shows the consistent gains compared to random selection using either low/high-selection ratio, ranging from 1/200 to 1/2. We also find the optimal sampling rate of a larger model (1B) is smaller than that of a smaller model (410M). We hypothesize that a larger model may require a more aggressive selection ratio, such as 10%, since it is more robust to the subtle distribution change of the training data. However, too low (1/200) or high selection ratio (1/2) does not perform as good, as a low ratio may harm the diversity and a high ratio does not leverage the strength of the data influence enough.

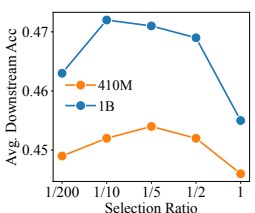

Figure 6: Zero-shot performances of MATES with different data selection ratios.

**Oracle Label Amount and Data Influence Model Scale.** Figure 7a demonstrates the impact of oracle label amounts by comparing the validation Spearman correlation of different data influence models at 40k steps. The data influence model is initialized from either the pretrained BERT or its last checkpoint at 30k steps. We observe that continuous fine-tuning from 30k steps requires less than half of the oracle data compared to training from the pretrained BERT, which can significantly reduce the cost of collecting new oracle data. Furthermore, Figure 7b studies the impact of parameter counts in the data influence model. Generally, the approximation becomes more accurate as the number of parameters increases until BERT-large, which may have become saturated with the current learning algorithm using the available oracle label. We have proved the effectiveness of our data influence models through extensive experiments and thus, leave the exploration of better data influence parameterization algorithms and stronger data influence models to future work.

### C.3 Analysis of oracle data influence

To further demonstrate the stability of our oracle data influence, we enlarge the one-step training batch size (BS) from 1 to 16. Following Ilyas et al. [26], we utilize LASSO regression to separate each data's influence score in the BS = 16 setup and calculate the Spearman correlation between BS = 1 and BS = 16 scores. More details of LASSO regression can be found in Appendix C.3. Figure 8a illustrates that the oracle data influence is not sensitive to the batch size as long as they correspond to the same model state. We calculate batch-level data influences, where one

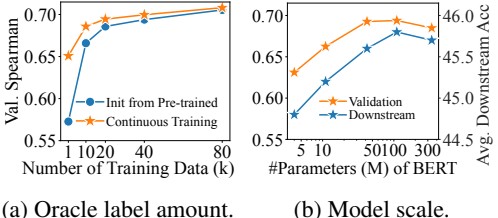

| (a) Oracle label amount. | (b) Model scale. |

Figure 7: Data influence parameterization with different amounts of oracle labels and model scales.

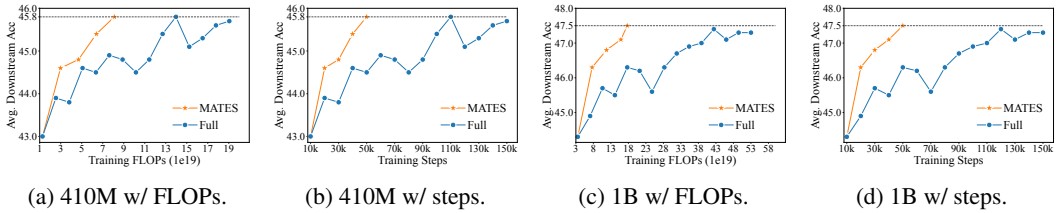

|  (a) 410M w/ FLOPs. | (b) 410M w/ steps. | (c) 1B w/ FLOPs. | (d) 1B w/ steps. |

Figure 9: Downstream performance of 410M and 1B models w.r.t. pretraining FLOPs and steps.

data point can appear on multiple batches so that separating individual data influence from batch influences is a regression problem. Formally, the input of the regressor is a $m$-dimensional binary vector $v : [v_1, v_2, ..., v_m]$ vector, where $m$ means the size of probing data pool (80k in our experiment) and $v_i = 1/v_i = 0$ denotes the inclusion/exclusion of the $i$-th data in the current batch; the output is the oracle data influence score probed by one-batch one-step training. In our experiment, we first collect 80k input-output pairs with batch size equal to 16 to train a LASSO regressor and then acquire a $m$-dimensional individual data influence vector from the regressor's prediction. This vector represents the influence score shown in the y-axis in Figure 8a. The idea of adopting LASSO regression is inspired by the linear datamodels in DsDm [15] that separate individual data influence from the collection of batch data influences.

We also investigate the traits of our reference task, LAMBADA, when collecting oracle data influences. Specifically, we collect the task accuracy at each model checkpoint and measure the Spearman correlation between a single task and average downstream accuracy. As shown in Figure 8b, LAMBADA has a significantly positive Spearman correlation (0.877) with average downstream accuracy. In contrast, the correlation between ARC-E/PIQA and average downstream accuracy is not high in Figure 8c/8d, although they are parts of the evaluation tasks. These results imply that ARC-E/PIQA may not reflect the target performance as accurately as LAMBADA when acting as the reference.

