# OpenReview forum: "MATES: Model-Aware Data Selection for Efficient Pretraining with Data Influence Models"
_NeurIPS.cc/2024/Conference — NeurIPS 2024 poster_

### Official Review · Reviewer_R3ZA · 2024-07-12

**Soundness:** 2
**Presentation:** 2
**Contribution:** 3
**Rating:** 5
**Confidence:** 4

**Summary:**

This paper introduces MATES, a method termed "model-aware data selection with data influence models". MATES is designed to optimize data selection for large language model (LLM) pre-training efficiently. The method dynamically considers various influence models during different stages of pre-training, tailored to the current state of the training model. Experiments validating this approach have been conducted using the C4 dataset with Pythia models.

**Strengths:**

- This paper reveals an important observation that the influence of data changes throughout the training process. It underscores the necessity for model-aware data selection methods to adapt accordingly.
- The utilization of the Gumbel-Top-k algorithm is particularly interesting and well-suited for balancing data quality and diversity. This diversity is crucial as it helps mitigate the independent influence assumption of the training data.
- Extensive experiments have been conducted to validate the method's effectiveness.

**Weaknesses:**

- **Gap between Motivation and Experiment Design**: While the authors have compared MATES with multiple baselines concerning data selection, there remains a gap regarding the necessity of data selection for pretraining as opposed to full training or no continuous pretraining.
  - **Full Training**: The results and computational costs of pretraining the entire C4 dataset have not been demonstrated. If the performance significantly drops or the computational cost reduction is minimal, then data selection may not be necessary initially.
  - **No Continuous Pretraining**: While MATES focuses on pretraining, the chosen model, Pythia, is already pretrained on the Pile dataset (i.e., pretraining a pretrained model). If Pythia already performs adequately on those tasks, then data selection might not be necessary initially.

- **Significance of the Results**: The results presented in Table 1 are not significantly robust. For instance, for Pythia-1B, only 5 out of 18 tasks show improvements greater than one **standard error**, suggesting that the p-values are likely higher than 0.05. In some recent similar works [1], the improvement over random selection can be up to seven times the **standard deviation**. Given that MATES also adds computational costs for influence computation, the significance of these results is arguable.

- **Missing Implementation Details**: Some critical details about the implementation are not sufficiently detailed, making the presentation somehow confusing. See the Question section.

[1] LESS: Selecting Influential Data for Targeted Instruction Tuning, ICML 2024.

**Questions:**

- For results presented in Table 1 and Figure 3, what is the amount of sampled data used for computing $I_m$? How to make the computation of $I_m$ efficient, considering that for each sampled data, one pass of the Adam and recomputation of the reference task loss are required?
- In relation to Figure 5(a), how is LASSO regression employed to "separate" influence scores? Does this imply that data points within the same batch can have varying influences?
- For Figures 4 and 6, what is implied by "Val. Spearman"? Was the influence computed for the entire dataset?

**Limitations:**

Besides the limitations discussed by the authors themselves, an additional limitation of MATES is that it uses a single pass of the Adam to approximate the change in loss. This approach assumes that each data point will be trained exactly once. While this assumption might be valid for certain pretraining scenarios, it does not hold for fine-tuning, where all data are used multiple times.

---

> ### Author Rebuttal · Authors · 2024-08-07
>
> Thank you for your review of our paper! We will address your questions/comments below:
>
> **Weakness 1:** Comparison with no continuous pretraining and full training
>
> **Response:** We first want to clarify that we only followed the Pythia architecture but **didn’t use any of its pretrained weights.** All the pretraining models in the paper are trained from scratch using C4. We chose C4 instead of the Pile since pretraining models on C4 can outperform the Pile, which has also been verified in the recent FineWeb paper. We will make this setup clearer in the next version.
>
> For the full training, we followed the original Pythia paper and pretrain the models using 150k steps. As shown in Figure 1 in our supplementary pdf, MATES can improve the pretraining efficiency by more than 3x, given that 50k MATES performance is comparable to or higher than the 150k full data training. Please see the general response 3.1 for more details.
>
> **Weakness 2:** Significance of the results
>
> **Response:** We hope to clarify that the **standard error** in Table 1 is computed across the accuracy of all the evaluation examples, **not the standard deviation** across different random seeds. Actually, this value is the “stderr” returned by the `lm-evaluation-harness`. It only shows the **variability** of the model’s accuracy across all the examples, **not the stability of each method**. We are aware that using $\pm$ to show this metric is inappropriate and will replace it with parentheses in the next version.
>
> LESS computes the standard deviation using different training seeds, while pretraining with different seeds is not feasible with our available computing resources. Our main baselines, DsDm and QuRating, also didn’t report the standard deviation across different seeds.
>
> We also notice that LESS and QuRating are papers from the same research group, while QuRating is mainly designed for pretraining data selection and LESS is for instruction tuning data selection. Generally, the average performance gain in pretraining is smaller than that in instruction tuning since pretrained models are typically evaluated on a broader set of benchmarks. Table 1 (both in our original paper and supplementary pdf) demonstrates that MATES **doubles the gain** achieved by the state-of-the-art method, QuRating, with fewer total FLOPs, so our improvement is really significant.
>
> Since LESS requires the computation of all training data’s gradients to perform the selection, our compute resources only allow us to run it at the short 40k decay stage (the same setup as Table 2). The results are shown below:
>
> |    | SciQ   | ARC-E  | ARC-C  | LogiQA  | OBQA   | BoolQ  | HellaSwag | PIQA   | WinoGrande | Average |
> | ----- | -------- | -------- | -------- | -------- | -------- | -------- | --------- | -------- | ---------- | -------- |
> | LESS | 63.0   | 41.3   | 23.6   | 24.0   | **28.8** | 54.6   | **42.1** | 64.4   | 51.3    | 43.7   |
> | MATES | **67.3** | **41.7** | **24.7** | **26.9** | **28.8** | **59.6** | 40.1   | **67.6** | **52.1**  | **45.4** |
>
> This shows that directly applying instruction data selection methods may not work well in the pretraining setup, suggesting that pretraining and instruction selections are two different research scenarios.
>
> **Question 1:** The amount of sampled data used for computing oracle scores and the way to make the computation efficient
>
> **Response:** The sampled data for computing oracle scores is 80k in the first round and 20k in the following rounds since we continuously fine-tune the data influence model. Even with the 1B model, the wall clock time to compute one oracle score is only 2.5s on one GPU, which means we can get all the required oracle scores during 50k-step pretraining on one node (8 GPUs) **around 12 hours**, which is significantly lower than the actual pretraining time (**4 days+**).
>
> To further make the influence computation efficient, we fine-tune a small data influence model to learn the oracle scores and predict the data influence. Even with the additional data selection costs, MATES greatly elevates the scaling curves (Figure 3) w.r.t. the total FLOPs, making this extra computation justifiable.
>
> **Question 2:** How is LASSO regression employed to separate influence scores? Does this imply that data points within the same batch can have varying influences?
>
> **Response:** Yes, data points within the same batch can have varying influences. We sample different batches to calculate the influence scores, and one data point can appear on multiple batches. Formally, the input of the regressor is a $m$-dimensional binary vector **$v: [v_1, v_2, ..., v_m]$** vector ($m$ means the total probing data pool, as mentioned in Algorithm 1), where $v_i=1$ denotes the inclusion of the $i$-th data in the current batch and vice versa; the output is the **oracle data influence score probed by one-batch one-step training**. We probed 80k input-output pairs that can be directly utilized to train the LASSO regressor, and we can get a $m$-dimensional individual data influence vector from the regressor’s weights. This method is inspired by the linear datamodels in DsDm [3] to separate the individual data influence from the collection of batch data influences. We will add more details about LASSO in the next version.
>
> **Question 3:** What is implied by "Val. Spearman"? Was the influence computed for the entire dataset?
>
> **Response:** The oracle influence is only computed on a tiny subset of 80k/20k examples (general response 2.2), not the entire. For the “Val. Spearman”, please see the general response 2.3 for more details.
>
> **Limitation 1:** Assumption that each data point will be trained exactly once does not apply for the fine-tuning.
>
> **Response:** Thank you for pointing out this. We assumed that the pretraining data is infinite w.r.t. current computing resources (line 26-27) and thus unique from other data selection scenarios, while most of the fine-tuning scenarios only have limited supervised examples.

---

> > ### Comment · Reviewer_R3ZA · 2024-08-12
> > **Thank you for your rebuttal**
> >
> > I appreciate the authors' comprehensive explanation and their efforts to address my previous concerns. The experiments conducted for full training are promising and effectively demonstrate the practical application of MATES. However, I suggest that for Figure 1 in the supplementary materials, it would be more accurate to label it as "Full" instead of "Random."
> >
> > Additionally, I recommend that the authors provide a detailed analysis of the computational overhead associated with data selection, including the training of the data influence model, in comparison to the pre-training costs. This would resolve the concern about the computational costs of this manuscript.
> >
> > While the approach of pretraining Pythia on C4 from scratch seems somewhat unconventional, I acknowledge that this paper opens up a promising direction for exploring how the influence of data evolves during different phases of pretraining. With the addition of the suggested clarifications, I believe this paper will make a valuable contribution to the community. Accordingly, I am inclined to raise my score to 5.

---

> ### Author Response · Authors · 2024-08-07
> **Code to compute our reported standard error**
>
> For your reference (Weakness 2), we provide the relevant code block to compute `stderr` from `lm-evaluation-harness`.
>
> ```python
> def sample_stddev(arr):
>
>   mu = mean(arr)
>
>   return math.sqrt(sum([(x - mu) ** 2 for x in arr]) / (len(arr) - 1))
>
> def mean_stderr(arr):
>
>   return sample_stddev(arr) / math.sqrt(len(arr))
> ```
>
> The `arr` in the code is passed with the binary accuracy list across all the examples, where 1 denotes model answers correctly and 0 denotes model answers wrong. The return value of `mean_stderr` only shows the variability of the model’s accuracy across all the examples, not across different random seeds as LESS did.

---

> ### Author Response · Authors · 2024-08-12
> **Thank you for your response**
>
> We appreciate your response and recognition of our work! For Figure 1 in our supplementary pdf, we will change the label name to be “Full”.
>
> For the computational overhead incurred by MATES, we reported the total pretraining FLOPs, including all the cost of maintaining the data influence model, in Table 1 in our original paper. Figure 3 also shows the total FLOPs-Acc curves, and MATES greatly elevates the scaling curves (even with additional data selection cost) compared to random selection. However, we agree that it is a good idea to have the detailed breakdown of the computational overhead incurred by MATES, so we provide it in the table below.
>
> | Pythia-410M                      | #FLOPs (x 1e19) | Ratio of the total #FLOPs |
> | -------------------------------- | --------------- | ------------------------- |
> | Main model pretraining           | 6.35            | 78.3%                     |
> | Oracle data influence collection | 0.30            | 3.7%                      |
> | Data influence model training    | 9e-3            | 0.1%                      |
> | Data influence model inference   | 1.46            | 18.0%                     |
> | Total                            | 8.11            | 100%                      |
>
>
> | Pythia-1B                        | #FLOPs (x 1e19) | Ratio of the total #FLOPs |
> | -------------------------------- | --------------- | ------------------------- |
> | Main model pretraining           | 17.67           | 88.5%                     |
> | Oracle data influence collection | 0.84            | 4.2%                      |
> | Data influence model training    | 9e-3            | 0.05%                     |
> | Data influence model inference   | 1.46            | 7.3%                      |
> | Total                            | 19.97           | 100%                      |
>
> The data selection cost of MATES only accounts for 21.8% (Pythia-410M) and 11.6% (Pythia-1B) of the total pretraining FLOPs, compared to the state-of-the-art method, QuRating, whose selection cost accounts for 75.9% (Pythia-410M) and 53.1% (Pythia-1B). The selection cost ratio of larger models is generally smaller since their pretraining cost dominates the total FLOPs while the training and the inference cost of our data influence model remains stable. The inference speed can also be improved with a fast-inference framework like vLLM [1]. We will include this new breakdown analysis in the next version of our paper.
>
> For the C4 dataset, we basically followed DsDm’s open-source version [2] to facilitate a fair and reproducible comparison. We also verified the effectiveness of MATES in the recent FineWeb [3] dataset (please refer to our general response 3.3), where MATES still demonstrates superior performance compared to the state-of-the-art data selection methods, such as the FineWeb-Edu classifier [3] and the fasttext-oh-eli5 classifier [4]. We will add this result in the next version.
>
> Thank you once again for your insightful reviews. Please do not hesitate to reach out if any part of our response requires further clarification.
>
> [1]: Kwon, Woosuk, et al. "Efficient memory management for large language model serving with pagedattention." Proceedings of the 29th Symposium on Operating Systems Principles. 2023.
>
> [2]: Engstrom, Logan, Axel Feldmann, and Aleksander Madry. "Dsdm: Model-aware dataset selection with datamodels." arXiv preprint arXiv:2401.12926 (2024).
>
> [3]: Penedo, Guilherme, et al. "The FineWeb Datasets: Decanting the Web for the Finest Text Data at Scale." arXiv preprint arXiv:2406.17557 (2024).
>
> [4]: Li, Jeffrey, et al. "DataComp-LM: In search of the next generation of training sets for language models." arXiv preprint arXiv:2406.11794 (2024).

---

### Official Review · Reviewer_T2sB · 2024-07-13

**Soundness:** 2
**Presentation:** 3
**Contribution:** 3
**Rating:** 5
**Confidence:** 3

**Summary:**

This paper introduces model-aware data selection with data influence models (MATES) that selects high-quality data for pre-training large language models (LLMs). MATES addresses the issue that existing data selection methods don’t adapt to evolving data preferences during pre-training. MATES continuously adapts to the main model’s evolving data preferences by uses a small data influence model to select the most effective pre-training data for the next stage. The data influence model also reduces the cost of computing the influence scores on-the-fly. Experiments on Pythia and the C4 dataset show that MATES outperforms random data selection and existing static data selection approaches. Further analysis validates the ever-changing data preferences of the pre-training models and the effectiveness of our data influence models to capture it.

**Strengths:**

1. The proposed method outperforms existing static data selection baselines on the Pythia-410 model. Compared to random selection, it can achieve a certain accuracy with less GPU FLOPS.
2. The authors demonstrate the importance of adapting to fluctuating data influence in data selection methods.

**Weaknesses:**

1. The authors only compare MATES to other static baseline methods on the Pythia-410M model. For most of the rest experiments, the authors only compare to random selection.
2. Missing references. There are existing dynamic data pruning methods [1][2] that the authors may want to include in the discussion of related works. These works also noticed the influence of data points changes as the training proceeds.

[1] Raju, Ravi S., Kyle Daruwalla, and Mikko Lipasti. "Accelerating deep learning with dynamic data pruning." arXiv preprint arXiv:2111.12621 (2021).
[2] Qin, Ziheng, et al. "InfoBatch: Lossless Training Speed Up by Unbiased Dynamic Data Pruning." The Twelfth International Conference on Learning Representations.

**Questions:**

1. The authors only compare against a full set of baseline methods in the Pythia-410M experiments. In the Pythia-1B experiments, the authors only compare against the random selection baseline. Why are the results of other baselines omitted in the Pythia-1B experiment? How do other baseline methods such as SemDedup perform in the Pythia 1B experiments?

2. How does the proposed method MATES perform compared to using the full dataset for training?

**Limitations:**

The authors adequately discuss the limitations of the work.

---

> ### Author Rebuttal · Authors · 2024-08-07
>
> Thank you for your review of our paper! We will address your questions/comments below:
>
> **Weakness & Question 1**: Only comparing MATES to other static baseline methods on the Pythia-410M model. For most of the experiments, the authors only compare random selection.
>
> **Response**: We acknowledge the reviewer’s concern regarding the limited comparison of MATES to other baselines, specifically for larger-scale models. It is important to note that large-scale experiments are costly and often infeasible for many university research groups. In research or even the productive large language models like T5 [1] and Llama 3.1 [2], it’s a common practice to conduct comprehensive ablations at a smaller scale and then test the final performance at large scales.
> With more resources, we are able to show all the 1B baselines in Table 1 in our supplementary pdf, where MATES still performs the best among all the baselines. This demonstrates our commitment to validating our approach at larger scales within our resource constraints, and we believe our results can provide valuable insights into MATES’s effectiveness across different model scales. Please see the general response 3.2 for more details.
>
> **Weakness 2**: Missing references.
>
> **Response**: Thanks for the references. Our work builds upon the growing emphasis on dynamic data selection for efficient model training. As with Raju et al. [3] and Qin. et al. [4], we share the common motivation of adapting the data selection as training progresses, as the value of training samples may change throughout the learning process.
> Unlike Raju et al.’s ε-greedy and upper confidence bound (UCB) approach, MATES uses a dedicated data influence model to capture the complex and changing relationships between training data and model performance. As a result, MATES is able to model more nuanced data preferences than methods based on simpler heuristics like the sum of mean and variance.
> Qin et al.’s InfoBatch maintains unbiased gradient expectations through gradient rescaling on the instruction fine-tuning setup. In MATES, we employ local probing to collect oracle data influence, providing a more direct measure of sample importance compared to InfoBatch’s loss-based soft pruning policy, allowing for a more accurate selection in the complex pretraining settings.
> We will cover these related works carefully in the next version.
>
> **Question 2**: How does the proposed method MATES perform compared to using the full dataset for training?
>
> **Response**: For the full training, we follow the original Pythia paper and pretrain the models with 150k steps. As shown in Figure 1 in our supplementary pdf, MATES can improve the pretraining efficiency by more than 3x, given that 50k MATES performance is comparable to or higher than the 150k full data training. Please see the general response 3.1 for more details.
>
> [1]: Raffel, Colin, et al. "Exploring the limits of transfer learning with a unified text-to-text transformer." Journal of machine learning research 21.140 (2020): 1-67.
>
> [2]: Dubey, Abhimanyu, et al. "The Llama 3 Herd of Models" arXiv preprint arXiv:2407.21783 (2024).
>
> [3] Raju, Ravi S., Kyle Daruwalla, and Mikko Lipasti. "Accelerating deep learning with dynamic data pruning." arXiv preprint arXiv:2111.12621 (2021).
>
> [4] Qin, Ziheng, et al. "InfoBatch: Lossless Training Speed Up by Unbiased Dynamic Data Pruning." The Twelfth International Conference on Learning Representations.

---

> > ### Comment · Reviewer_T2sB · 2024-08-11
> >
> > I appreciate the authors for conducting additional experiments and providing the response. The response answers my question about missing references and full dataset training. However, according to the additional 1B zero-shot results provided in Table 1 of the newly uploaded pdf, the proposed MATES wins the best baselines on 3 out of the 10 datasets, draws with the best baselines on 2 out of the 10 datasets, and loses to the best baselines on 5 out of the 10 datasets. This seems to be worse compared to the 410M zero-shot results (wins on 7 / 10, loses on 3 / 10). Does this suggest the advantage of MATES degrades as the model size increases? I will adjust my score if this question is clear.

---

> ### Author Response · Authors · 2024-08-11
>
> Thank you for your detailed review and follow-up questions. We believe the reduced margin on 1B is because our selection ratio and candidate data pool are not optimal for the 1B setting. We don't have enough compute resources to intensively tune these hyperparameters since our work is at an exploratory scale, as mentioned in our limitations section.
>
> For the selection ratio, we **fix it to 20%** in our main table. However, if we vary it from **20% to 10%** for 1B at the 50k decay stage, we can observe some gains (also shown in Figure 3a in our supplementary pdf):
>
> |      | SciQ     | ARC-E    | ARC-C    | LogiQA   | OBQA     | BoolQ    | HellaSwag | PIQA     | WinoGrande | Average  |
> | :--- | :------- | :------- | :------- | :------- | :------- | :------- | :-------- | :------- | :--------- | :------- |
> | 10%  | **67.8** | 44.4     | 25.5     | **28.9** | **32.6** | **60.9** | **47.4**  | **70.5** | **52.4**   | **47.8** |
> | 20%  | 67.3     | **44.9** | **25.9** | 28.7     | 32.2     | **60.9** | 45.3      | 69.5     | **52.4**   | 47.5     |
>
> With the 10% ratio, MATES (Pythia-1B) can have a 6/0/4 #Win/#Tie/#Loss over the best baselines. This indicates that we still have headroom to improve MATES with a more suitable selection ratio in the full training run (not only at the decay stage due to the rebuttal time limit).
>
> For the selection pool, we **fix it to 125B tokens for all models**, but we believe a larger pool will generally benefit 1B models. Recent DataComp-LM [1] also shows that the improvement trends observed on a smaller scale (410M) align with the larger scale (1B and 7B) **when the larger models have a larger selection pool (e.g., 15T tokens at most)**. We plan to verify the effectiveness of MATES on this standard DataComp-LM benchmark and will update the results with a more suitable ratio and pool in the next version.
>
> Additionally, for your reference, we report the #Win/#Tie/#Loss of all our baseline methods in their original papers:
>
> | Method   | Baselines                          | #Win/#Tie/#Loss over the best baseline as reported in their papers |
> | -------- | ---------------------------------- | ------------------------------------------------------------ |
> | DSIR     | Random, Manual, Heuristics         | 4/0/5                                                        |
> | SemDeDup | Random, NearDup                    | 2/0/1                                                        |
> | DsDm     | Random, Classifier, DSIR, SemDeDup | 5/1/9                                                        |
> | QuRating | Random, DSIR                       | 5/0/6; If we only compare MATES with Random and DSIR, the #Win/#Tie/#Loss is 8/1/1 |
>
> Note that all these baseline papers have already been accepted at the top-tier conferences. This underscores the point that it is reasonable for an effective method to not outperform the best baseline in every task but rather to achieve a solid average performance gain across tasks. We recognize that each method has its unique strengths. For instance, QuRating is specifically optimized for educational data and thus is likely to significantly elevate the model performance on the knowledge QA tasks. On the other hand, MATES stands out due to achieving the highest overall performance while utilizing few additional FLOPs.
>
> Thank you once again for raising this question. Please do not hesitate to reach out if any part of our response requires further clarification.
>
> [1]: Li, Jeffrey, et al. "DataComp-LM: In search of the next generation of training sets for language models." arXiv preprint arXiv:2406.11794 (2024).

---

> > ### Comment · Reviewer_T2sB · 2024-08-14
> >
> > I appreciated the authors' response. I will increase the score.

---

### Official Review · Reviewer_xrwH · 2024-07-13

**Soundness:** 3
**Presentation:** 2
**Contribution:** 3
**Rating:** 5
**Confidence:** 5

**Summary:**

It is important to carefully select data during the pretraining stage, as the pretraining data are often obtained from web crawling and can be extensive and noisy. Existing methods for data selection include heuristic-based approaches, clustering-based techniques, and the use of influence models. However, these methods result in static selected datasets and do not adapt to the evolving training process. This paper introduces a novel approach by developing an influence model that co-evolves with the main pretraining model. The selected data is no longer static; now, it is dynamic and sensitive to the model. The study demonstrates that continuously adapting data preferences enables efficient training, and the influence model is both small and accurate. Empirical evidence presented in the framework, MATES, proves its better performance compared to other methods, showcasing advantages in both performance and efficiency.

**Strengths:**

* Originality: This paper introduces a new data selection framework in the pretraining stage by learning a smaller data influence model and co-evolving with the main model's data preference.
* Quality: Empirically, their results show promise to enable efficient and improved pretraining.
* Clarity: Their writing and their result are clear to me.
* Significance: Data curation is crucial to denoise massive pretraining datasets. This paper presents a new approach to selecting data points by learning model preferences. Their empirical results support their claim, and I believe their notion is significant to the field.

**Weaknesses:**

1. In Figure 2, it appears that the data influence model needs to be reinitialized at each iteration. It's not clear why we need to reset it instead of continuing to learn with the main model.
2. It's unclear how the influence model includes and excludes data points and converts them into labels for learning the influence score. Could the authors clarify this? Additionally, what is the performance of the influence model across training epochs and how do we evaluate it? I believe there is an imbalance in the labels.
3. Although the learning curve indicates efficient and improved performance, it's uncertain how much computational burden is involved in maintaining the trained influence model. It would be helpful to include these details.

**Questions:**

1. It's an interesting paper, and I am curious about the changes in data preference. Can the authors provide some examples of what the model likes to learn at the beginning but discards later, or vice versa?
2. This is not critical: but looking forward to seeing this approach being employed in other data curation benchmarks, such as DataComp.

**Limitations:**

I didn't see any potential negative societal impact of their work.

---

> ### Author Rebuttal · Authors · 2024-08-07
>
> Thank you for your review of our paper! We will address your questions/comments below:
>
> **Weakness 1**: The initialization of the data influence model.
>
> **Response**: We only initialize our data influence model with pretrained BERT at 10k steps and continuously fine-tune it at the following steps. Figure 6a shows that initializing the data influence model with pretrained weights still works well with enough training data (80k), while continuous fine-tuning demands less training data to achieve the same performance, saving the oracle probing cost by 75% (80k -> 20k).
>
> **Weakness 2**: Unclear how the influence model includes and excludes data points and converts them into labels for learning the influence score.
>
> **Response**: The oracle data influence score is defined as the difference in the target loss before and after one-step training on a hold-out example $x_i$ (Equation 4). Please see the general response 2.2 for more details.
>
> **Weakness 3**: What is the performance of the influence model across training epochs, and how do we evaluate it? Is there an imbalance in the labels?
>
> **Response**: We already reported the performance of data influence models during pretraining in Figure 4. Please see the general response 2.3 for more details.
>
> The oracle data influence distribution conforms nearly to the normal distribution, so there is no imbalance in the labels (Figure 3b in our supplementary pdf). As training steps increase, our data influence model can better capture the data influence of the main model with increasing validation performance.
>
> **Weakness 4**: The computational burden in maintaining the trained influence model.
>
> **Response**: As stated in lines 198-199, we reported the total pretraining FLOPs, including the cost of maintaining the data influence model, in Table 1. The actual additional data selection cost incurred by the MATES is (8.11 - 6.35) * 10^19 = **1.76 * 10^19** FLOPs for the 410M model and (19.97 - 17.67) * 10^19 = **2.3 * 10^19** FLOPs for the 1B model, which is near 9x lower than the selection cost of the state-of-the-art data selection method QuRating (**20 * 10^19** FLOPs). Figure 3 also reports the FLOPs-ACC curves and MATES greatly elevates the scaling curves (even with additional data selection cost) compared to random selection.
>
> **Question 1**: Some examples of what the model likes to learn at the beginning but discards later, or vice versa?
>
> **Response**:  We would like to present a table of the high-influence cases on a hold-out 500k subset whose rank decreases a lot across the training.
>
> | Selected Data Rank (Lower means higher Influence) | Source                | Text                                                         |
> | -------------------------------------------------- | --------------------- | ------------------------------------------------------------ |
> | 0 in 10k checkpoint; 6376 in 20k checkpoint                             | Blog                  | She went to the place where Jonathan lay and gave to his servant's David's richest garment to be placed next to him as he lay crying out in his sickness. She went in and out of the house. She went in and out of the city gates. She waited for David in the place |
> | 1 in 20k checkpoint; 22998 in 30k checkpoint                             | Wikipedia             | Two weeks later, Friedman threw three touchdown passes in a 27–0 victory over Northwestern. One of Michigan's touchdowns was set up when Friedman intercepted a Northwestern pass and returned it 13 yards. |
> | 3 in 30k checkpoint; 452 in 40k checkpoint                              | Slides in CRITHINKEDU | Critical Thinking Across the European Higher Education Curricula), Education and Regional Development in Southern Europe: Should we invest in Critical Thinking across the Higher Education Curricula? First… What is Critical Thinking (CT)? CT is not only a high quality way of thinking (skill), but also a way of being (disposition). |
> | 1 in 40k checkpoint; 5954 in 10k checkpoint                            | forkliftcertification | It’s a group of training course resources to help you master telescopic forklifts in record time. Or else you’re taking the course and throwing a bunch of forklift telescopic training against a wall and hoping something sticks. And forklift is only getting more popular. This chapter is about handler course and certification. |
>
> We find that the model at the early pretraining stage (10k steps) prefers to learn the natural narrative w/o too much specific knowledge. In the 20k steps, the model focuses more on factual knowledge (e.g., the page from Wikipedia). In the 30k checkpoint, the model prefers more academic text, such as the official teaching slides. In 40k steps, the model turns to more long-tail knowledge like Telescopic forklift Training. This study shows that the model’s preferences for the data continuously evolve during the pretraining. We will add this analysis in the next version of the paper.
>
> **Question 2**: Employ MATES on other data curation benchmarks, such as DataComp.
>
> **Response**: We agree that DataComp [1]  is a good future experiment. In another recently released state-of-the-art pretraining data FineWeb [2] (Figure 2 in our supplementary pdf), MATES still demonstrates better average performance compared with the strong FineWeb-Edu classifier (similar to QuRating but using annotations generated by LLama3-70B-Instruct) and fasttext-oh-eli5 classifier (the best-performing classifier to mine instruction-like data in the DataComp paper). This result gives us confidence that MATES can contribute to the top-performing runs on the DataComp leaderboard under its standardized data selection setup.
>
> [1]: Li, Jeffrey, et al. "DataComp-LM: In search of the next generation of training sets for language models." arXiv preprint arXiv:2406.11794 (2024).
>
> [2]: Penedo, Guilherme, et al. "The FineWeb Datasets: Decanting the Web for the Finest Text Data at Scale." arXiv preprint arXiv:2406.17557 (2024).

---

> > ### Comment · Reviewer_xrwH · 2024-08-12
> >
> > Thanks for providing additional experiments and clearing my concerns about initialization, evaluation, and computation. Results in your attachment look promising. Thanks for providing examples that evolve with model preference. They are interesting and love to see more follow-up exploration.
> >
> > Overall, I think this paper is interesting and shows good empirical results, but some places lack detailed descriptions. I will increase my score to 5. Thanks for your response!

---

> > > ### Author Response · Authors · 2024-08-12
> > >
> > > Thank you very much for your response and recognition of our work! We will clarify the experimental setup and add detailed descriptions about our method and results in the next version.

---

### Official Review · Reviewer_2Zbt · 2024-07-16

**Soundness:** 3
**Presentation:** 2
**Contribution:** 3
**Rating:** 6
**Confidence:** 5

**Summary:**

The paper proposes a new method “MATES,” which aims to select pretraining data using a reference high-quality data source. MATES uses an estimated influence function to iteratively select the most influential datapoints. Experiments on the Pythia model + dataset show good promise over existing data curation methodologies.

**Strengths:**

- Better trained models on MATES’ curated data vs. other data curation approaches
- The distillation of influence scores into a smaller proxy model (BERT-base in the paper) allows MATES to perform iterative, model-state dependent data selection vs. other baselines which perform one-step, model-state-independent data selection.
- Thorough experiments: various experiments and ablations are included, e.g., various relevant baseline comparisons, ablating the effect of various hyper-parameters in MATES, etc.

**Weaknesses:**

- I found the paper writing to be quite loose and missing context in various places. I had to read a lot of the text multiple times as well as filling in the missing context myself.
- While I appreciate the extensive experiments covered in the paper, I feel the paper misses a few important results and/or ablations. See the questions section for more on this.
- The choice of evaluation benchmarks doesn’t seem obvious and feels quite debatable to me. See the questions section for more on this.

**Questions:**

While I completely understand the cost of experiments included in the paper, I believe a few critical questions/extra experiments might greatly improve the paper. Please feel free to ignore any of the following questions in the interest of compute.


- I couldn’t find details on which dataset was used. While the abstract mentions C4, the main text doesn’t specify this, making me question if the Pythia dataset was used instead?
- The above question also makes way for a deeper question: since the selection ratio is fixed to 20% (of the entire dataset, I assume), this translates to how many *tokens sampled*? Are the training runs multi-epoch or is the sampled data greater than the training budget?
- If the runs are multi-epoch, what are the results in the full-data (no sampling) case?
- How does MATES compare to other approaches when the sampling rate is not equal to 20%? Is MATES also useful in the super low/high-sampling rate regime?
- What if a different target dataset is used vs. only using lambada? Can we use a mixture of target datasets? How does the design of the target dataset/mixture affect different evals?
- Finally, the evaluation benchmarks used in the paper seem to be cherry-picked based on the “high-level” proximity to the kind of data present in lambada. The paper would greatly benefit from using more diverse evaluation scenarios like coding, math, knowledge, etc.

**Limitations:**

The authors address the major technical limitations of their work in Section 6. No obvious negative societal impact.

---

> ### Author Rebuttal · Authors · 2024-08-07
>
> Thank you for your review of our paper! We will address your questions/comments below:
>
> **Weakness 1:** Loose writing and missing context.
>
> **Response:** Thank you for pointing it out! We will carefully revise the paper from the following perspectives:
>
> 1. Clarify the experimental setup, as mentioned in general response 2
> 2. Add more theoretical derivation from the classic influence function formulae [1] to the representation of our oracle data influence
> 3. Avoid grammar errors and missing context and add more detailed explanations for both our method and results
>
> **Question 1:** Pretraining dataset used.
>
> **Response**: We use the C4 dataset (the same as DsDm) only to pretrain all our main models from scratch. We will clarify it in the next version of the paper.
>
> **Question 2:** Can the selection ratio translate to the number of tokens? Are the training runs multi-epoch, or is the sampled data greater than the training budget?
>
> **Response:** The ratio can be translated into the number of tokens. In every 10k steps, we update the data influence model and select the data from separate pools randomly sampled from the entire C4. Each data selection pool is 5x larger than the training data of 10k steps, so we only select 20% from it. The whole pretraining data pool (C4) is huge, so we run one epoch training as most previous work (DsDm, QuRating) did.
>
> **Question 3:** Full-data results?
>
> **Response:** For the full training, we follow the original Pythia paper and pretrain the models with 150k steps. As shown in Figure 1 in our supplementary pdf, MATES can improve the pretraining efficiency by more than 3x, given that 50k MATES performance is comparable to or higher than the 150k full data training. Please see the general response 3.1 for more details.
>
> **Question 4:** How does MATES compare to other approaches when the sampling rate is not equal to 20%? Is MATES also useful in the super low/high-sampling rate regime?
>
> **Response:** Interesting question! We only have time to provide ablation results of different selection ratios in Figure 3a in our supplementary pdf, in which MATES shows consistent gains compared to random selection using either low/high-selection ratio, ranging from 1/200 to 1/2. We also find the optimal sampling rate of a larger model (1B) is smaller than that of a smaller model (410M). However, too low (1/200) or high selection ratio (1/2) will decrease the performance, as a low ratio may harm the diversity, and a high ratio does not leverage the strength of the data influence enough.
>
> For your reference, this is the adopted selection ratio from our baseline papers. Our optimal selection ratio is close to QuRating/DsDm, which pretrains a model of a similar size as ours.
>
> | Methods   | Max Model Size    | Selection Ratio |
> | ----------- | ---------------- | --------------- |
> | DSIR    | 110M | 3.2%      |
> | SemDeDup  | 6.7B | 63%       |
> | QuRating  | 1.3B | 11.5%      |
> | DsDm    | 1.3B | 20%       |
> | DataComp-LM | 6.9B  | 10%       |
>
> **Question 5:** A different or a mixture of target datasets? How does the design of the target dataset/mixture affect different evals?
>
> **Response:** Thanks for your suggestion. We use FLAN (the mixture of multiple NLP tasks) as the target dataset to run our MATES on the recently released FineWeb datasets [2]. As shown in Figure 2a in our supplementary pdf, MATES still demonstrates superior average downstream performance on FineWeb compared to the state-of-the-art data selection methods, such as the FineWeb-Edu classifier and the fasttext-oh-eli5 classifier. Compared with taking LAMBADA as the target task, using FLAN can improve the model’s performance on both knowledge QA and commonsense tasks (Figure 2c, Figure 2d). We hypothesize that multi-task data (e.g., FLAN) has better coverage of the language task types and thus is more suitable for serving as the target task. Please see the general response 3.3 for more details.
>
> **Question 6:** Choice of the evaluation tasks
>
> **Response:** We followed the original Pythia paper and our main baseline QuRating for the evaluation benchmark selection and didn’t cherry-pick any tasks that resemble LAMBADA. LAMBADA is essentially a word prediction task and greatly differs from our QA-like evaluation tasks. It is found to be a great reference task due to its high correlation (0.877) with the average downstream accuracy (Figure 5b) and the high data influence modeling (>0.7) performance (Figure 6a).
>
> We agree that coding, math, and knowledge are essential to evaluate a language model’s capabilities. For the knowledge part, actually most of our adopted QA tasks (SciQ, ARC-E, ARC-C, OBQA) require factual and commonsense knowledge to solve. In our supplementary pdf, we also evaluate MMLU [3] in Figure 2b. On MMLU, MATES can outperform random selection but not the FineWeb-Edu classifier. As mentioned in general response 3.3, the FineWeb-Edu classifier is optimized towards educational values derived from LLama3-70B-Instruct, which can benefit knowledge-intensive tasks like MMLU with the price of losing generalization abilities to other types of evaluation tasks. For the coding and math tasks, we didn’t include them since they are not commonly used to evaluate pretrained checkpoints w/o additional instruction tuning. A supportive evidence is that even the Llama 3 series [4] **didn’t choose to evaluate pretrained checkpoints** on HumanEval (coding) and MATH (math) due to the non-indicative performance.
>
> [1]: Koh, Pang Wei, and Percy Liang. "Understanding black-box predictions via influence functions." Proceedings of the 34th ICML-Volume 70. 2017.
>
> [2]: Penedo, Guilherme, et al. "The FineWeb Datasets: Decanting the Web for the Finest Text Data at Scale." arXiv preprint arXiv:2406.17557 (2024).
>
> [3]: Hendrycks, Dan, et al. "Measuring massive multitask language understanding." arXiv preprint arXiv:2009.03300 (2020).
>
> [4]: Dubey, Abhimanyu, et al. "The Llama 3 Herd of Models" arXiv preprint arXiv:2407.21783 (2024).

---

> ### Author Response · Authors · 2024-08-13
> **Looking Forward to Your Reply**
>
> Dear Reviewer 2Zbt,
>
> We have carefully addressed your feedback in our rebuttals and provided detailed responses to each of your comments. We believe these clarifications will help in assessing our work more comprehensively.
>
> We would greatly appreciate it if you could review our rebuttals and provide any further feedback, given that the author-reviewer discussion will be closed on Aug. 13 at 11:59 p.m. AoE in no more than one day. We are willing to answer any further questions.
>
> Thank you for your time and consideration. We look forward to your reply.
>
> Best,
>
> The Authors

---

### Author Rebuttal · Authors · 2024-08-07

## 0 Overview

First, we thank all the reviewers for their great efforts and insightful feedback.

In this post, we summarize positive points from the reviews, clarify the experimental setup, and address the shared questions proposed by the reviewers with additional experiments to support and strengthen our findings.

## 1 Positive points

We sincerely thank all the reviewers for recognizing our key contributions. We appreciate the acknowledgment of our novel data selection framework MATES, which learns a smaller data influence model that co-evolves with the main model’s data preference to perform iterative, model-state dependent data selection ( `2Zbt`, `xrwH`). Furthermore, we are grateful for their recognition of our method’s superior performance: MATES underscores the necessity for model-aware data selection in pretraining, outperforming static baselines with fewer GPU FLOPs (`2Zbt`, `xrwH`, `T2sB`, `R3ZA`). We are also pleased that our extensive and varied experiments and ablations were noted for validating the method’s effectiveness (`2Zbt`, `R3ZA`).

## 2 Clarification of the setup

We summarize the most important clarifications and discuss each in detail in the following points:

1. **The pretraining dataset used in this work (`2Zbt` and `R3ZA`):** We use the **C4** dataset only to pretrain all our main models from **scratch.**

1. **The training data of the data influence model (`xrwH` and `R3ZA`):** The oracle data influence score is defined as **the difference in the target loss** **before and after one-step training** (i.e., $\mathcal{L}(\mathcal{D_t} \mid \mathcal{A}(\mathcal{M}, x\mid_{i \to -1}))$ and $\mathcal{L}(\mathcal{D_t} \mid \mathcal{A}(\mathcal{M}, x\mid_{i \to +1}))$ in Equation 4) using the Adam optimizer on a hold-out example $x_i$. Then, we construct the mapping of the data point to its influence score $(x_i, \mathcal{I}_\mathcal{M}(x_i;\mathcal{D}_t))$ (line 148) as the training data for our BERT-based data influence model. The size of sampled data for computing data influence scores (i.e., $m$ in Algorithm 1) is **80k** at 10k steps and **20k** at 20k, 30k, 40k, 50k steps.

1. **The evaluation of the data influence model (`xrwH` and `R3ZA`):** We evaluate our data influence model’s performance by its prediction’s Spearman correlation with the oracle data influence on the **10% hold-out validation set** sampled from the collected data influence training set {$(x_i, \mathcal{I}_\mathcal{M}(x_i;\mathcal{D}_t)) \mid x_i \in S_m$} (Algorithm 1). Spearman correlation is an intermediate metric and a higher correlation denotes the better learning of our data influence models that approximate the oracle scores (Figure 4a). Our ultimate evaluation of the data influence model is still the downstream performance of the main model pretrained with the selected data (Figure 4b).

We will clarify all of those in the next version of the paper.

## 3 Additional experiments

We summarize observations of the additional experiments we have provided in the supplementary pdf below:

1. **Pythia-410M/1B full data training run (`2Zbt`, `T2sB`, and `R3ZA`), in Figure 1:** We provide the full data training experiments (following the original Pythia pretraining steps of 150k) of Pythia-410M/1B in Figure 1. For the full data training curve, we notice that the average downstream performance increases very slowly, while MATES significantly elevates the model performance in the early pretraining steps. MATES can improve the pretraining efficiency by more than 3x, given that 50k MATES performance is comparable to or higher than the 150k full data training.
2. **All baselines in the Pythia-1B (`T2sB`), in Table 1:** We are able to secure more compute resources to run all the 1B baseline results and present those in Table 1. Among all the baselines, MATES still performs the best in both zero-shot and two-shot performances. We would like to highlight that zero-shot evaluation is the more important and robust one to reflect a model's pretraining abilities, while two-shot evaluation depends on the in-context examples used, which can be less stable.
4. **MATES on FineWeb [1] with LAMBADA or FLAN as target tasks (`2Zbt`), in Figure 2:** To verify MATES’s effectiveness on a more advanced dataset, we use LAMBADA or FLAN [2] (the mixture of multiple NLP tasks) as the target dataset to run MATES on FineWeb. FineWeb is the state-of-the-art open-source pretraining dataset released by HuggingFace recently. It is processed with complicated and delicate data curation steps. We also include two new baselines in the FineWeb comparison: one is the FineWeb-Edu classifier that is similar to QuRating and the LLM quality filter in LLama3.1, using educational scores generated by LLama3-70B-Instruct; the other is fasttext-oh-eli5 classifier that mines instruction-like data from DataComp corpus and achieves the best selection results in the DataComp paper [3]. As shown in Figure 2a, MATES demonstrates superior average downstream performance on FineWeb compared to these state-of-the-art data selection methods. Using multi-task FLAN as the target can make MATES generalize better to downstream tasks than using LAMBADA only. In contrast, the FineWeb-Edu classifier overfits the knowledge QA tasks (Figure 2c) but underperforms in commonsense tasks (Figure 2d), as the scores it assigns to data are influenced by human-written criteria that may disproportionately favor certain downstream capabilities.

We will add and analyze all of those in the next version of the paper.

[1]: Penedo, Guilherme, et al. "The FineWeb Datasets: Decanting the Web for the Finest Text Data at Scale." arXiv preprint arXiv:2406.17557 (2024).

[2]: Chung, Hyung Won, et al. "Scaling instruction-finetuned language models." Journal of Machine Learning Research 25.70 (2024): 1-53.

[3]: Li, Jeffrey, et al. "DataComp-LM: In search of the next generation of training sets for language models." arXiv preprint arXiv:2406.11794 (2024).

---

> ### Author Response · Authors · 2024-08-07
> **TL;DR of the setup clarification**
>
> | Name                                   | Value                                                        |
> | -------------------------------------- | ------------------------------------------------------------ |
> | Pretraining dataset                    | C4, trained from scratch                                     |
> | Amount of collected oracle data        | 80k at 10k steps and 20k at 20k, 30k, 40k, 50k steps         |
> | Initialization of data influence model | Initialized with pretrained BERT at 10k steps and continuously fine-tuned at 20k, 30k, 40k, 50k steps |
> | Validation set of data influence model | 10% sampled from the collected oracle data                   |

---

### Decision · Program_Chairs · 2024-09-25

**Decision:**

Accept (poster)

**Comment:**

Reviews are somewhat borderline, though certainly leaning on the side of accepting since all lean positive. Reviewers praise the originality of the work, and the findings themselves, as well as aspects of the data/experiments, etc. Arguably somewhat "generic" praise though still uniformly positive. Some issues raised about the writing, evaluation, and various other small things, though these seemed fairly minor.